# Pupil size predicts exploration through critical slowing in prefrontal dynamics
Akram Shourkeshti[1,5], Mojtaba Abbaszadeh [1,5], Gabriel Marrocco[1], Katarzyna Jurewicz[1,2], Tirin Moore [3,4] & R. Becket Ebitz [1]✉

In uncertain environments, intelligent decision-makers exploit actions that have been rewarding in the past, but also explore actions that could be better. Several studies link exploration to pupil size—a peripheral correlate of neuromodulatory tone and arousal. However, pupil size may only track variables that make exploration more likely, such as volatility or reward, without directly predicting exploration or its neural bases. Here, we simultaneously measured pupil size, exploration, and neural population activity in the prefrontal cortex while two male rhesus macaques explored and exploited in a dynamic environment. We find that pupil size under constant luminance specifically predicts the onset of exploration beyond effects of reward history. Pupil size also predicts disorganized patterns of prefrontal activity at the single neuron and population levels. Our results support a model in which pupil-linked mechanisms drive exploration by pushing prefrontal dynamics through a critical tipping point.

Many decisions maximize immediate rewards. However, in uncertain or changing environments, it is important to sacrifice some immediate rewards in order to learn about the value of alternative options and discover new, more valuable strategies for interacting with the world. In short, in complex environments, intelligent decision-makers exploit rewarding strategies, but also explore alternative strategies that could be even better.

Because exploitation maximizes immediate reward, it can rely on the same value-based decision-making processes that have been the subject of neurobiological studies for decades[1–5]. However, we are only just beginning to understand the neural bases of exploration[6–10]. One clue is that many organisms seem to explore via random sampling[9–11]. Randomness is a critical component of exploratory discovery in bird song and motor learning[12,13], it can perform about as well as more sophisticated strategies in many environments[14], and humans and other primates tend to explore randomly even when more sophisticated strategies are available[9,15]. There is some neurobiological evidence linking random exploration to disorganized activity patterns in the prefrontal cortex[10,15–17], but we still do not understand the proximate causes of random exploration and its neural correlates in the prefrontal cortex.

One promising hypothesis is that exploration could be under the control of some process(es) linked to pupil size. Pupil size under constant luminance is a peripheral index of autonomic arousal[18–20] that also predicts widespread changes in neural population activity[21,22]–including in regions implicated in decision-making noise[23,24]. Among other neuromodulators[25–27], pupil size is correlated with central norepinephrine[28,29]: a catecholamine that flattens neuronal tuning functions[30] and predicts "resets" in cortical networks[31,32]. Behaviorally, pupil size predicts decision-making noise[10,25,31,33–35], especially errors of reward-maximization[36] and task performance[23,33]. Some of these "errors" may be caused by exploratory processes[16,36,37].

There is a plausible alternative interpretation of this data: perhaps pupil size only tracks the variables that make exploration more likely. Pupil size under constant luminance increases with the volatility of reward environments, the surprise of reward outcomes, novelty, uncertainty, and context changes[38–43]: all variables that make exploration more likely. However, it is often unclear whether the pupil is tracking these variables or instead directly predicting behavioral changes, such as increased learning, decision-noise or exploration[35,44,45]. Fortunately, recent results suggest that at least some exploration appears to occur tonically, regardless of these variables[10,16,37]. Further, in parallel, new computational approaches allows us to determine when exploration is occurring independently of the reward-based computations thought to drive it[15,16,46,47]. This means that it is now possible to determine whether pupil size predicts exploration itself or instead simply tracks the variables that make exploration more likely.

Here, we measured pupil size and recorded from populations of prefrontal neurons while two rhesus macaques performed a task that encouraged exploration and exploitation. Rhesus macaques were chosen because their prefrontal cortex and decision-making mechanisms closely parallel those of humans[48], making them a suitable model for investigating the

[1]Department of Neurosciences, Université de Montréal, Montréal, QC, Canada. [2]Department of Physiology, McGill University, Montréal, QC, Canada. [3]Department of Neurobiology, Stanford University School of Medicine, Stanford, CA, USA. [4]Howard Hughes Medical Institute, Chevy Chase, MD, USA. [5]These authors contributed equally: Akram Shourkeshti, Mojtaba Abbaszadeh. ✉e-mail: becket@ebitzlab.com

neural basis of exploration and cognitive control. We found that pupil size under constant luminance was larger during explore choices than exploit choices. However, the temporal relationship between pupil size and exploration was both precise and complex: spontaneous oscillations in pupil size entrained the onset of exploration. Together, these results support the hypothesis that pupil-linked processes drive the prefrontal cortex through a critical tipping point that permits exploratory decisions.

## Results

Two male rhesus macaques performed a total of 28 sessions of a classic explore/exploit task: a restless three-armed bandit (subject B: 10 sessions, subject O: 18 sessions; a total of 21,793 trials). Some analyses of this dataset have been reported previously[15], but the pupil data has not been analyzed previously and all analyses presented here are new. In this task, the reward probability (value) of three targets walks randomly and independently over time (Fig. 1A). This means that the subjects have to take advantage of valuable options when they are available (exploit), but also occasionally sample alternative options to determine if they have become more valuable (explore).

Rather than instructing subjects to explore and exploit, this task takes advantage of the subjects' natural tendency to alternate between exploration and exploitation in a changing environment. We have previously shown that both monkeys and mice exhibit 2 behavioral modes in this task: one exploitative mode in which they repeatedly choose the same option—learning little but maximizing reward—and one exploratory mode in which they alternate rapidly between the options—choosing randomly with respect to rewards and learning rapidly[15,46]. We infer which of these modes is driving behavior with a hidden Markov model (HMM; Fig. 1B; see Methods). This approach models the exploratory and exploitative modes as latent goal states and the maximum a posteriori goal is taken as the state label for each choice. We have previously shown that this method identifies explore/exploit state labels that match normative definitions[15,46] and explain variance in prefrontal neural activity that cannot be explained by reward value, reward history, and switch/stay decisions[15]. This task design naturally elicits exploration and exploitation, allowing us to investigate variability in pupil size and neural activity under both conditions.

Some previous studies used a different method to identify exploratory choices[7,8,36]. These studies fit a reinforcement learning (RL) model to the behavior and identified the choices that are not consistent with the model's subjective values as exploratory. However, this previous RL-based approach (1) equates exploration with errors of reward maximization, not a goal that is orthogonal to reward maximization, and (2) its accuracy depends on precise knowledge of the computations involved in the choice, which are highly variable, both across individuals and over time[46,49]. The HMM approach, conversely, makes no assumptions about the computations involved in the choice and identifies choices that are orthogonal to reward value, not anti-correlated with it[15,46]. Here, we found that state labels from the HMM method explained more variance in behavior and neural activity than choice labels from the previous, RL method (Fig. 1C; response time: both subjects, paired t-test: $p < 0.005$, t(27) = 3.41, the mean difference of beta weights = 0.004, 95% CI = 0.002–0.007, Cohen's d = 0.64, $n = 28$; scatter index[15]: both subjects, paired t-test: $p < 0.001$, t(27) = 3.84, the mean difference of beta weights = 0.15, 95% CI = 0.07–0.24, Cohen's d = 0.73, $n = 28$; see Methods). In short, we find that the HMM approach is a more robust and accurate method, with better face validity, than the RL-based method for identifying explore choices. Therefore, here, we used this more precise approach to determine whether physiological signals, like pupil size, reliably track exploratory behavior.

### Pupil size is larger during exploratory states

Having established that the HMM reliably distinguishes explore and exploit states, we next asked whether pupil size changes across these behavioral states. Previous work using RL-based labels reported that pupil size under constant luminance is larger during exploration than exploitation[36]. We therefore tested whether this pattern holds using HMM-based labels in our dataset. Indeed, we found that pupil size at fixation (see Methods) was larger

on explore-labeled trials than exploit-labeled trials in both subjects (Fig. 1D; both subjects, paired t-test: $p < 0.0001$, t(27) = 4.95, mean offset = 0.23, 95% CI = 0.13–0.32, Cohen d = 0.94, $n = 28$; subject B: $p < 0.001$, t(9) = 5.50, mean offset = 0.4, 95% CI = 0.24–0.57, Cohen d = 1.74, $n = 10$; subject O: $p < 0.02$, t(17) = 2.85, mean offset = 0.13, 95% CI = 0.03–0.23, Cohen d = 0.67, $n = 18$). Thus, pupil size was larger during exploratory choices identified with the HMM method.

However, the probability of exploration did not increase linearly as a function of pupil size (Fig. 1E). A linear, first-order GLM confirmed that larger pupil size generally predicted more explore choices (both subjects: $\beta = 0.084$, $p < 0.0001$, 95% CI = 0.059–0.110, AIC = −1053.70, $n = 28$ sessions). This relationship held when analyzed separately in each subject (subject B: $\beta = 0.063$, $p = 0.002$, 95% CI = 0.023 to 0.103, AIC = −368.01, $n = 10$ sessions; subject O: $\beta = 0.110$, $p < 0.0001$, 95% CI = 0.068–0.152, AIC = −249.46, $n = 18$ sessions). Yet the relationship was clearly non-linear. A quadratic model provided a significantly better fit than the linear model for the combined dataset (2nd order GLM: $\beta_1 = -0.081$, $p = 0.101$, 95% CI = −0.179–0.016; $\beta_2 = 0.166$, $p = 0.0006$, 95% CI = −0.072–0.260; AIC = −1063.62, AIC weight for the quadratic model = 0.993), consistent with a U-shaped relationship. This U-shape was especially prominent in subject O ($\beta_1 = -0.13$, $p = 0.090$, 95% CI = −0.278–0.018; $\beta_2 = 0.240$, $p = 0.001$, 95% CI = −0.097–0.384; AIC = −258.01), although the quadratic model was not an improvement over the linear model in subject B ($\beta_1 = -0.027$, $p = 0.709$, 95% CI = −0.174–0.118; $\beta_2 = -0.091$, $p = 0.205$, 95% CI = -0.049–0.231; AIC = -367.63).

In order to determine whether this pattern was also apparent in raw switching probability (i.e., not the HMM-model labels), we next asked if pupil size predicted choices to a different option than the previous trial. We again observed a U-shaped relationship between pupil size and the probability of making a switch choice in both subjects (1st-order GLM: $\beta = 0.084$, $p < 0.0001$, 95% CI = 0.062 to 0.107, $n = 28$ sessions; subject B: $\beta = 0.0800$, $p < 0.0001$, 95% CI = 0.047 to 0.112; subject O: $\beta = 0.0807$, $p < 0.0001$, 95% CI = 0.049–0.112). A 2nd-order quadratic model provided a superior fit in both animals (both subjects: $\beta_1 = -0.099$, $p = 0.023$, 95% CI = −0.184 to −0.013; $\beta_2 = 0.184$, $p < 0.0001$, 95% CI = 0.101 to 0.266; subject B: $\beta_1 = -0.069$, $p = 0.284$, 95% CI = −0.187 to 0.048; $\beta_2 = 0.149$, $p = 0.010$, 95% CI = 0.036 to 0.263; subject O: $\beta_1 = -0.093$, $p = 0.103$, 95% CI = -0.204 to 0.017; $\beta_2 = 0.174$, $p = 0.001$, 95% CI = 0.066 to 0.281). Model comparison strongly favored the quadratic model (linear AIC = −1195.05; quadratic AIC = −1211.95; AIC weight for quadratic model = 0.9998). Thus, although pupil size tended to be larger during exploration than exploitation, its relationship with both exploration and switching was clearly U-shaped.

One possible explanation for the U-shaped pattern is that some "explore" choices—particularly those with small pupil size—reflect disengagement at low levels of arousal, rather than true exploration. However, if this were the case, then the valid, large-pupil explore choices would systematically differ from the false, small-pupil "explore" choices. To evaluate this possibility, we compared small- and large-pupil explore trials across 5 key behavioral and neural dimensions that we have previously shown differentiate exploration from exploitation[15,46,48]: reward rate, saccade velocity, the neural scatter index, a trial-wise learning index, and reaction time. Small- and large-pupil explore choices (median split) were essentially indistinguishable along each of the key dimensions (Fig. 1F). For example, both were equally likely to be rewarded (mean difference = 0.03 ± 0.24 STD) between large- and small pupil-explore choices ($p > 0.4$, t(1,27) = 0.75, Cohen's d = 0.142, paired t-test; AUC for discriminating explore and exploit = 0.65 ± 0.05 STD across sessions). Both had similar peak saccadic velocities (mean difference = −0.05 ± 0.23 STD, $p > 0.2$, t(27) = -1.08, Cohen's d = 0.20; explore/exploit AUC = 0.61 ± 0.10 STD) and both had more variability in neural population choice information ("scatter index", mean difference = 0.03 ± 0.33 STD, $p > 0.6$, t(27) = 0.45, Cohen's d = 0.09; explore/exploit AUC = 0.60 ± 0.07 STD). Both had similar levels of reward learning (see Methods; the mean difference = -0.03 ± 0.57 STD, $p > 0.7$, t(27) = 0.27, Cohen's d = 0.05): in both cases, learning was substantially enhanced relative to the exploit choices (small-pupil, the mean difference

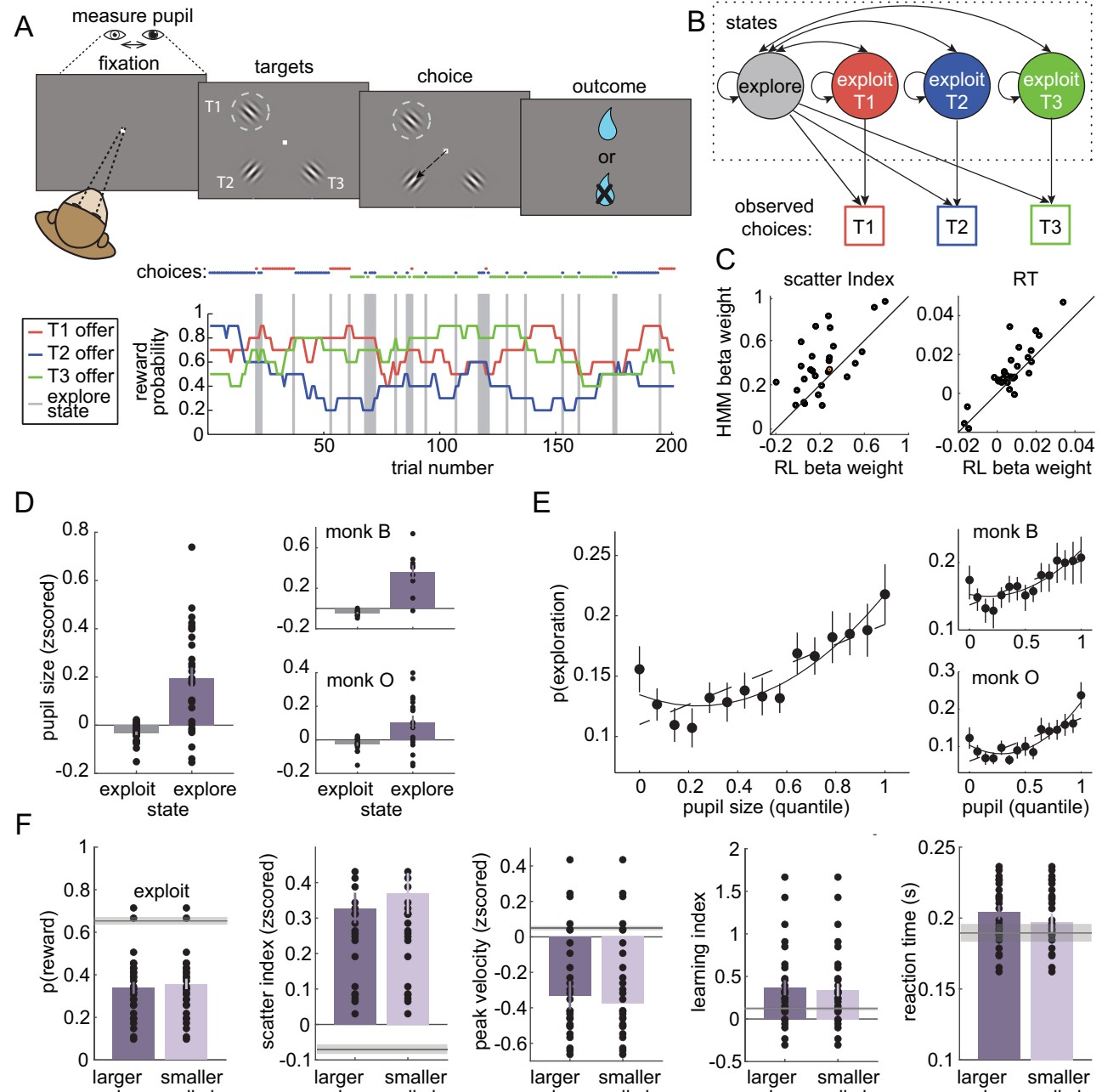

**Fig. 1 | Task design and pupil. A** Top: Subjects made saccadic choices between three identical options (T1, T2, and T3). One of the options (e.g., T1 in this example trial) was located in the receptive field of a neuron in the frontal eye field (FEF; dotted circle). Bottom: Reward probabilities for the three options (lines), with choices overlaid (dots) for 200 example trials. Gray bars = explore-labels. **B** The HMM models exploration and exploitation as latent goal states underlying choice sequences. **C** Comparison of regression coefficients for HMM-inferred and RL-inferred explore choices, predicting either the disorganization of neural population responses ("scatter index"; see Methods) or response time. Separate models were fit using either the explore labels from the HMM or from an RL model. **D** Average pupil size on explore and exploit choices. Right: Same for individual subjects. Dots show individual data points for each session, and the overlaid solid lines indicate the SEM. **E** The probability of explore choices as a function of pupil size quantile. Dotted line: linear GLM fit. Solid line: quadratic fit. Right: Same for individual subjects. **F** Several behavior measures compared across median-split large- and small-pupil-size explore choices. Left to right: reward probability, a one-trial-back learning index (see Methods), saccadic peak velocity of saccades, the scatter index, and reaction time. No significant differences between pupil bins. Dots show individual data points for each session, and the overlaid solid lines indicate the SEM. The gray line is the mean ± SEM for exploit choices. Error bars depict ± SEM throughout.

from exploit = 0.24 ± 0.48 STD, $p < 0.02$, $t(27) = 2.69$, Cohen's $d = 0.51$; large-pupil, the mean difference from exploit = 0.21 ± 0.39 STD, $p < 0.01$, $t(27) = 2.91$, Cohen's $d = 0.55$). Reaction times were also similar across small- and large-pupil explore choices (mean difference = 0.01 ± 0.02 STD, $p > 0.6$, $t(27) = 1.84$, Cohen's $d = 0.35$; explore/exploit AUC = 0.58 ± 0.06 STD). These results are incompatible with the idea that either type of explore choice reflects disengagement in the task or that small- and large-pupil

explore choices have different causes. Instead, we will see that the U-shape was due to the complex temporal relationship between pupil size and exploration.

**Pupil size, but not other measures ramp up before exploration**
Pupil size ramped up across trials before exploration began in both subjects. After exploration, it shrank to below-baseline levels when exploitation

resumed (Fig. 2A). Here, "baseline" refers to a z-scored value of 0, computed by subtracting the session mean and dividing by the session standard deviation of pupil size (see Methods).This ramping meant that pupil size was larger not just during exploration, but also during the exploit choices immediately before exploration (both subjects, GLM slope ($\beta$) = 0.015, $p = 0.002$, 95% CI = 0.005–0.025, n = 28; subject B: $\beta = 0.02$, $p = 0.021$, 95% CI = 0.003–0.036, $n = 10$; subject O: $\beta = 0.013$, $p = 0.044$, 95% CI = 0.0004–0.025, $n = 18$; average pupil size compared to the exploit choices, post-hoc paired t-tests, 1 trial before exploration mean = 0.12, $p = 0.002$, t(27) = 3.416, Cohen's d = 0.64; 2 trials mean = 0.09, p = 0.023, t(27) = 2.41, Cohen's d = 0.45; 3 trials mean = 0.03, $p = 0.166$, t(27) = 1.42, Cohen's d = 0.26; 4 trials mean = 0.05, p = 0.047, t(27) = 2.073, Cohen's d = 0.39). By the first exploit choice after exploration, pupil size had already begun shrinking to below-baseline levels (post-hoc paired t-tests, 1 trial after exploration mean = 0.03, $p = 0.09$, t(27) = 1.733, Cohen's d = 0.32; 2 trials after mean = -0.11, $p = 0.012$, t(27) = -2.669, Cohen's d = -0.51; 3 trials after mean = -0.16, $p = 0.011$, t(27) = -2.721, Cohen's d = -0.51; 4 trials after mean = -0.08, $p = 0.215$, t(27) = -1.268, Cohen's d = -0.23; 5 trials after mean = -0.16, $p < 0.001$, t(27) = -3.941, Cohen's d = -0.74; p-values are significant with a Holm-Bonferroni correction). The shrinking to below-baseline levels could suggest a refractory mechanism that would prevent exploration from re-occurring immediately after it happened.

To rule out potential confounds, we tested whether the pupil ramping and shrinking effects could be explained by misaligned labels or unrelated behavioral dynamics. We saw no evidence of ramping in peak saccadic velocity, another behavioral measure that differentiated explore trials and exploit trials (Fig. 2B; no significant decrease from baseline one trial before, paired t-test: $p > 0.7$, t(27) = 0.42, Cohen's d = 0.07; a GLM was non-significant with the trend pointing in the opposite direction: 15 trials preceding exploration, beta = 0.003, $p > 0.3$) and no significant change from baseline afterward (not greater than the baseline during the five trials after exploration, when pupil shrinking was maximal, mean = -0.47 ± 0.15 STD, $p > 0.9$, one-sided t(27) = -1.67, Cohen's d = -0.31). This null result resonates with results seen in neural data in prior work[15]. Thus, while pupil size ramped before exploration began and shrank afterward, the same was not true of other behavioral and neural variables, suggesting that these dynamics were not some artifact of misalignment.

## Pupil size generally ramps across trials, but resets with exploration

To better understand whether pupil ramping was a general feature of arousal dynamics or specific to exploration, we examined how pupil size evolved across trials with and without exploratory transitions (see Methods). When the subjects did not explore the pupil size increased steadily across trials (Fig. 2C; both subjects, GLM: beta = 0.004, $p < 0.0001$, 95% 0.004–0.005; subject B: beta = 0.003, $p < 0.0001$, 95% CI = 0.002–0.005; subject O: beta = 0.005, $p < 0.0001$, 95% CI = 0.004–0.006, $n = 25$ lags over 28 sessions). This implies that the ramping in pupil size before explore choices may be a general dynamic of how pupil size evolves in the absence of exploration. However, a different pattern emerged when we looked at how the pupil changed between exploit trials that were separated by exploration. When two exploit trials were separated by at least one explore choice, pupil size was smaller on the second exploit trial (both subjects, GLM: beta = -0.07, $p < 0.0001$, 95% CI = -0.1 to -0.05; subject B: beta = -0.12, $p < 0.0001$, 95% CI = -0.16 to -0.08; subject O: beta = -0.05, $p < 0.0001$, 95% CI = -0.08 to -0.02). Therefore, pupil size tended to ramp across trials but exploratory choices temporarily decreased pupil size without disrupting this ramping in the long term.

## Pupil size specifically predicts the onset—not the maintenance—of exploration

Pupil size tends to be smaller after exploration, but this shrinkage could either be driven by the end of exploration (i.e. the start of exploitation) or it could begin shortly after the beginning of exploration itself. If the pupil starts to shrink only after exploration ends, it would support models that suggest that

pupil size decreases with commitment to a new option or belief state[35]. Conversely, if the pupil shrinks immediately after exploration begins, it might suggest that pupil-linked mechanisms are important for initiating exploration, but not sustaining it. Our results were consistent with the latter hypothesis: the pupil immediately began shrinking as soon as exploration began, not when it ended (Fig. 2D; mean change in pupil size between neighboring explore choices = -0.17, t-test, t(27) = -2.69, $p < 0.02$; 95% CI = -0.30 to -0.04). This was essentially identical to the magnitude with which the pupil shrank on exploit trials that followed explore trials (mean change = -0.17, t-test, t(27) = -2.56, $p < 0.02$; 95% CI = -0.31 to -0.03). Validating the ramping we observed with other methods, we also found that pupil size tended to grow on explore trials that followed exploit trials here (mean change = 0.14, t-test, t(27) = 2.96, $p < 0.01$; 95% CI = 0.04 to 0.24). Together, these results suggest that pupil size and pupil-linked mechanisms specifically predict the "onset" of exploration—the first exploratory trial in a sequence—and may not be important for sustaining exploration after the first explore choice.

The tendency of the pupil to shrink after the onset of exploration could explain the previously noted U-shaped relationship between pupil size and exploration. In this view, the small-pupil-size explore choices would be the later explore choices in a sequence and the larger pupil size explore choices would tend to be the first explore choice(s). Indeed, pupil size had a primarily linear relationship with the onset of exploration in both subjects (Fig. 2E; 1st order GLM: $\beta = 0.042$, $p < 0.0001$, 95% CI = 0.031–0.053, AIC = –1973.57, $n = 28$). Adding a quadratic term did not substantially improve the model fit ($\beta_2 = 0.038$, $p = 0.073$, 95% CI = 0.003–0.080; quadratic model AIC = –1974.81; $\Delta$AIC = –1.24; AIC weight of quadratic model = 0.65; see Methods). This linear relationship was also observed in both monkeys individually (see Fig. 2E, right panels). For subject B, a first order GLM confirmed a significant positive association ($\beta = 0.044$, $p < 0.0001$, 95% CI = 0.028–0.060, AIC = –699.13, $n = 10$), and adding a quadratic term did not improve the model fit ($\beta_2 = 0.049$, $p = 0.0878$, 95% CI = 0.006 to 0.104; $\Delta$AIC = –0.96). Similarly, for subject O, pupil size showed a significant linear relationship with exploration onset ($\beta = 0.034$, $p < 0.0001$, 95% CI = 0.019–0.049, AIC = –460.29, n = 18), and the quadratic model again provided no additional explanatory power ($\beta_2 = 0.022$, $p = 0.402$, 95% CI = -0.030–0.075; $\Delta$AIC = +1.30). These results confirm that the linear relationship between pupil size and the onset of exploration was robust across both subjects and not driven by outliers or subject-specific variability. Conversely, there was no special relationship between pupil size and probability of starting to exploit (1st order GLM: beta = 0.05, $p > 0.05$). Thus, pupil size specifically predicted the onset of exploration, rather than explore choices or state switches more generally.

If the U-shaped relationship between pupil size and exploration (Fig. 1E) were driven primarily by later explore trials, it should remain evident after excluding onset trials from the analysis. Moreover, removing onsets should substantially reduce the slope of the linear effect. To test this, we repeated the analysis using only later explore trials. As expected, the linear slope decreased in both subjects (Supplementary Fig. 1): for subject B, $\beta_1$ dropped from 0.063 (all explore trials) to 0.018 (excluding onsets), and for subject O, from 0.110 to 0.075. In contrast, the quadratic terms remained relatively stable: for subject B, $\beta_2$ was 0.091 for all explore trials and 0.041 with onsets excluded; for subject O, $\beta_2$ was 0.240 and 0.215, respectively. Importantly, the U-shaped relationship persisted when data were combined across both subjects (Supplementary Fig. 1), with the quadratic model significantly outperforming the linear model (AIC quadratic = –1108.84, linear = –1103.30; relative AIC weight for the quadratic model = 0.941). These findings confirm that the non-linearity observed in the original analysis (Fig. 1E) was driven by the decrease in pupil size in later exploratory trials, whereas the onset of exploration had a largely linear relationship with pupil size.

## Exploration is gated by pupil-linked arousal, not just reward history

Although pupil size predicted the onset of exploration, it remained possible that this relationship was driven by a shared sensitivity to recent reward

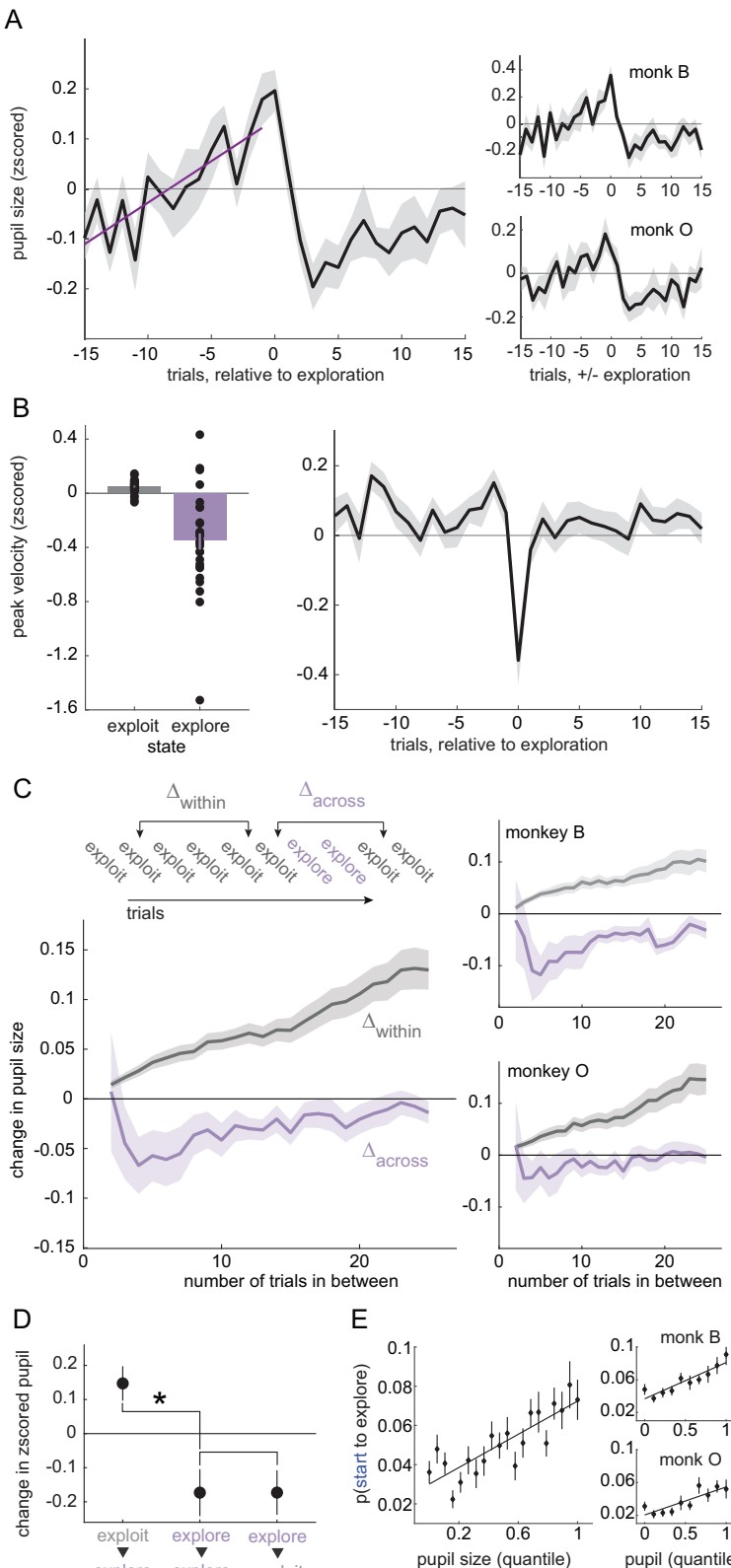

**Fig. 2 | Pupil size ramps up before exploration and shrinks down after. A** Average pupil size for 15 trials before and 15 trials after explore choices. Purple line: GLM fit. Right: Same for each subject separately. **B** Same as Figs. 1D and 2A but for peak velocity rather than pupil size. Dots show individual data points for each session. **C** Change in pupil size between exploit trials that are either in a single bout of exploitation (gray) or separated by explore trials (purple). Right: Same for each subject separately. **D** Change in pupil size over certain pairs of trials: starting (exploit to explore), during (explore to explore), and leaving (explore to exploit) exploration. *$p < 0.001$ **E** The probability of starting to explore as a function of pupil size quantile. Solid line: Linear GLM fit. Error bars and shaded regions depict mean ± SEM. Insets: Same analysis shown separately for each monkey. Error bars and shaded areas depict ± SEM throughout.

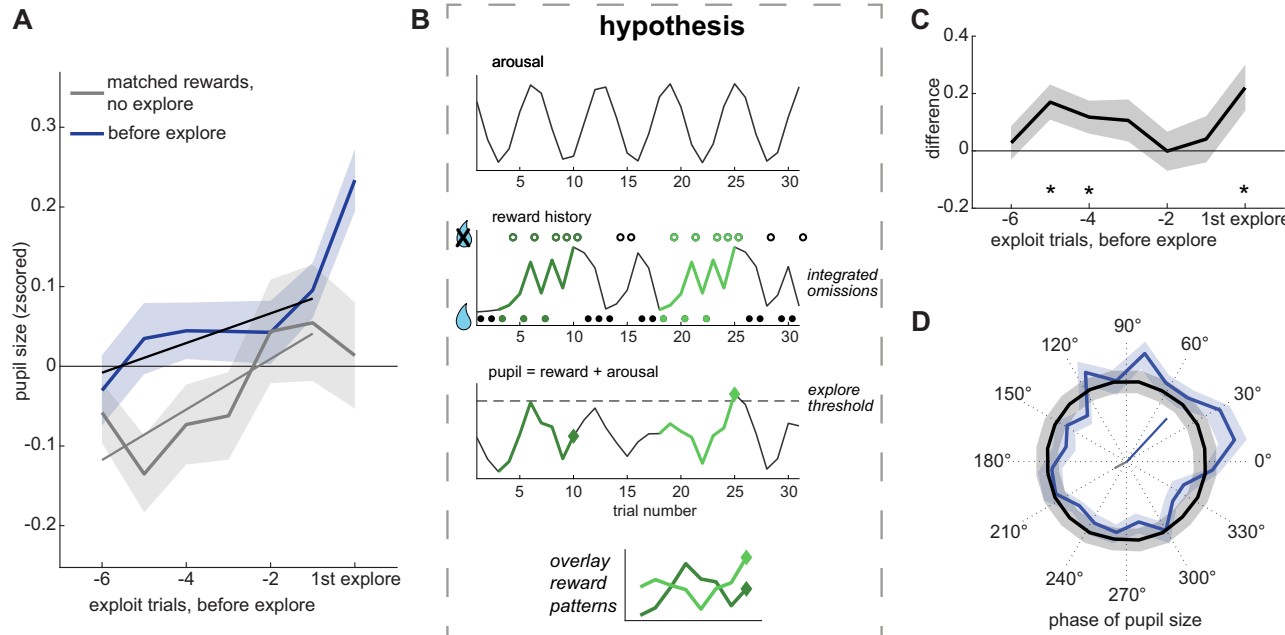

**Fig. 3 | The onset of exploration is phase-locked to pupil size. A** Average pupil size over sequences of exploit trials before the onset of exploration (black line) and sequences with matched rewards, but no exploration at the end (gray line). Lines: GLM fit. **B** Cartoon illustrating how oscillations in arousal (top) could interact with reward history (middle), to regulate exploration. The bottom panel illustrates a hypothetical pupil trace that has an additive effect of reward omissions and by oscillating arousal. Exploration (diamond shapes) begins when pupil size reaches a threshold (dotted line). Note that identical patterns of reward delivery and omission have different outcomes, depending on how they align with the phase of arousal (gray = no exploration, blue = exploration). **C** Difference in pupil size between the traces in (**A**). **D** Phase distribution of pupil size at the onset of exploration (blue) and bootstrapped null distribution (black). The vectors at the center indicate the mean vector direction and length for the trials before exploration (blue) and the matched reward trials (gray). Shaded areas ± SEM throughout.

outcomes, since both exploration[7,9,15] and pupil dilation[36,50] tend to increase following reward omission. To determine if there was a direct effect of pupil-related processes on exploration, we compared pupil size across exploit trials before exploration with pupil size from matched trial sequences where exploration did not happen (see Methods). There was a significant increase in pupil size during the trials before exploration compared to "matched rewards" control trials (Fig. 3B; GLM, beta = 0.025, $p < 0.01$, n = 28), suggesting that pupil size predicted the onset of exploration beyond what could be explained by reward. Again, pupil size ramped up over time (GLM, beta = 0.119, $p < 0.02$, $n = 28$), but this ramping did not differ between the traces (GLM, beta = 0.007, $p > 0.5$, $n = 28$). This implies that either reward history or time (i.e., the number of trials) may explain the pupil ramping before exploration, although there is still an offset in pupil size that predicts the onset of exploration above and beyond the effect of reward history.

Visual inspection of Fig. 3A suggested that there may be a phase difference in pupil size between trials where exploration began and matched reward trials where exploration did not begin. This led us to develop a hypothesis (Fig. 3B): that rewards may interact with ongoing oscillations in pupil size. Due to delays in communication between the baroreceptor reflect and changes in heart rate, the sympathetic nervous system[51–56] has a natural oscillation known as the Mayer wave, with a period of approximately 0.05–0.1 Hz[51,53]. Critically, transitions in other behavioral states can be entrained by this oscillation, with eliciting stimuli causing transitions only at certain phases of arousal. If pupil size is out of phase between trials that lead to exploration and trials that do not, then the previous trials most predictive of the onset of exploration may actually be several trials prior to onset itself—during the periods in which out-of-phase signals would be most distinct. Indeed, the trials in which pupil size best predicted the onset of exploration were not those immediately preceding it (e.g., trial t–1 or t–2), but rather trials t–4 and t–5 (Fig. 3C; trial t–4, mean difference = 0.117, $p < 0.05$, t(27) = 2.09; trial t–5 = 0.170, $p < 0.01$, t(27) = 2.84).

The view that omitted rewards only evoke exploration when they coincide with particular phases of sympathetic arousal also implies that that

the onset of exploration should be phase-locked to the Mayer wave frequency (see Methods). The median trial duration was ~3 seconds (range = [2.2, 3.2]), so a ~ 5-trial cycle would correspond to a 0.06–0.09 Hz oscillation, which aligns with the frequency range of the Mayer wave. We found that pupil phase at the onset of exploration was concentrated at the rising phase (Fig. 3D; mean phase = 47.18°, Hodges-Ajne test, p < 0.01; vector length = 0.075; null = 0.026, 95% CI = 0.004–0.057, $p < 0.0001$). In contrast, pupil phases during reward-matched trials pointed in the opposite direction (mean phase = 207.58°; significantly different from onsets, p < 0.02, Watson's U² = 0.25, $n = 2170$ phases including 1135 onsets). Together, these results support the hypothesis (Fig. 3B) that slow, rhythmic fluctuations in arousal interact with reward history to determine the timing of exploration onset.

## Pupil size predicts flattened neural tuning in prefrontal cortex during exploration

To probe the neural mechanisms linking pupil size to exploration, we examined how pupil size predicts neural activity in the FEF[57–60] (Fig. 4A). We previously reported that exploration is associated with flattened tuning for choice in FEF neurons[15]. While FEF neurons often predict upcoming choices during exploitation, many show reduced choice selectivity during exploration. Pupil size predicted similar changes in FEF neurons and did so beyond what could be explained by exploratory states themselves. Out of 155 recorded single neurons, 88 (57%) were tuned for choice (Fig. 4B; 57%, one sample proportion test: $p < 0.001$). These are referred to as "tuned neurons," regardless of whether they were modulated by pupil size. Among tuned neurons, 21 (24%) were also modulated by pupil size, and 16 (18%) showed a significant interaction between choice and pupil size. On average, tuning curves flattened as pupil size increased in both tuned and untuned neurons (Fig. 4C, D). Among untuned neurons, an additional 22% (15/67) were significantly modulated by pupil size ($p < 0.05$), with a median regression coefficient (β) of –0.0011 ± 0.063. This suggests that pupil-linked mechanisms affect FEF activity even in neurons that are not directly

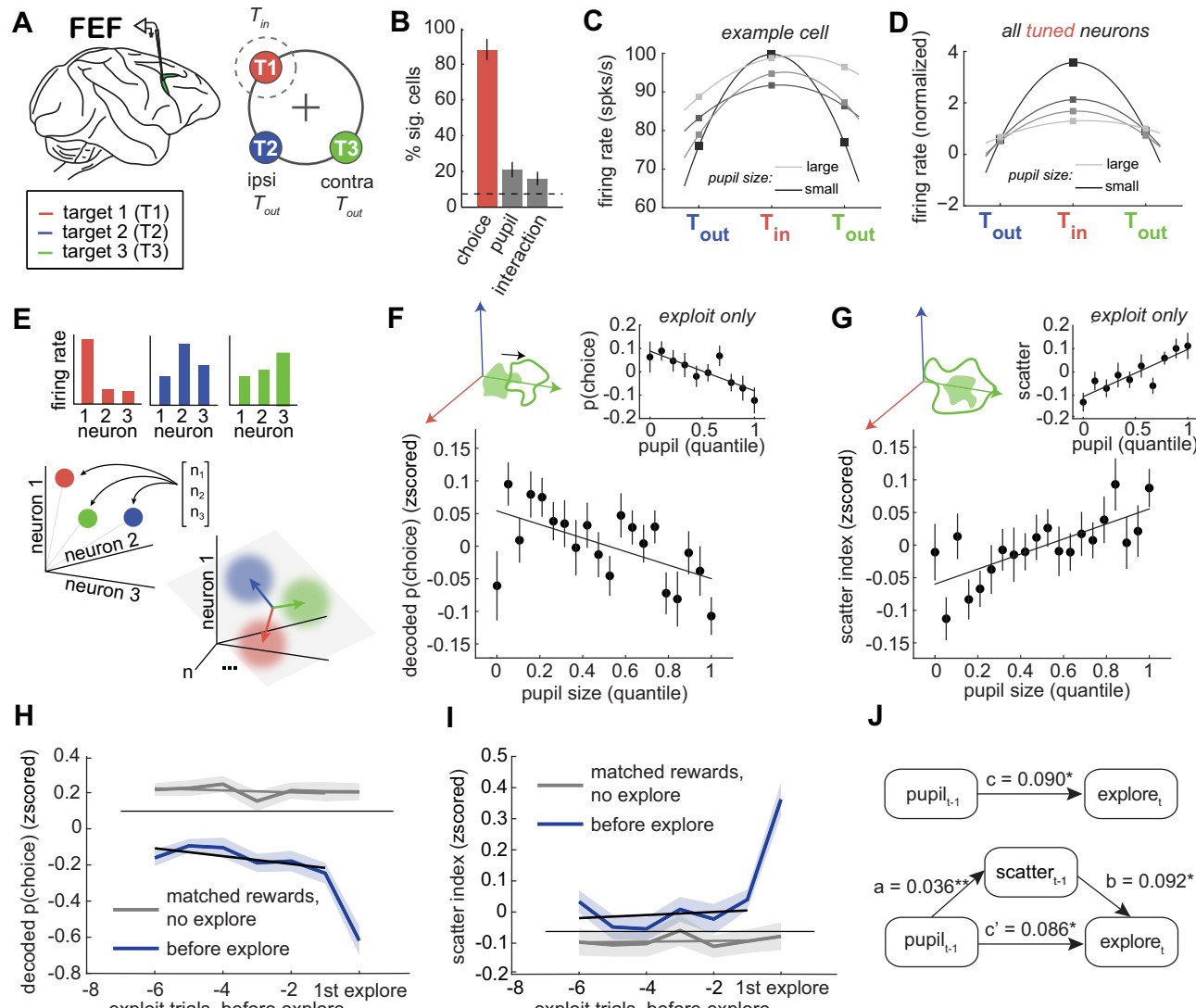

**Fig. 4 | Pupil size predicts choice tuning curves and population disorganization.**
**A** Recordings were made in the FEF. Right: The cartoon illustrates the relative positions of the receptive field target ($T_{in}$, red) and the ipsilateral and contralateral targets ($T_{out}$, blue and green). **B** Percent of neurons with significant tuning for choice target, pupil size, and the interaction. **C** Tuning curve for an example neuron across target locations, separated by pupil size. Lighter = larger pupil. The dashed line indicates the number of neurons expected to be significant by chance at $p < 0.05$. **D** Same for all tuned neurons, which refers to the 88 out of 155 FEF neurons that were significantly tuned for choice, regardless of their modulation by pupil size. **E** Cartoon illustrating how neural population measures consider patterns of firing rates across neurons as vectors in neural state space. Targeted dimensionality reduction is used to find the hyperplane where the distribution of neural activity

across trials best predicts choice. Vectors here are the coding dimensions that separate the choices. **F** The decoded choice probability (projection onto the correct coding dimension) plotted as a function of pupil size quantile. Inset: Same for exploit trials alone. **G** The scatter index, a measure of the variance in choice-predictive population activity, plotted as a function of pupil size quantile. Inset: Same for exploit trials. **H** Decoded choice probability for trials before the onset of exploration (in blue) and trials with matched rewards (in gray). **I** Scatter index for trials before the onset of exploration (in blue) and trials with matched rewards (in gray).
**J** Mediation analysis between pupil size, scatter index, and the onset of exploration. Top: Direct model. Bottom: Indirect, mediated model. Asterisks marked significant paths (*$p < 0.01$ **$p < 0.001$). Error bars and shaded areas depict ± SEM throughout.

involved in encoding choice. This may suggest a more domain-general role for arousal in modulating prefrontal network dynamics.

Because single neurons are noisy, it is difficult to dissociate the effects of pupil size and exploration at the level of individual cells. However, changes in tuning at the single-neuron level also imply shifts in the organization of the neural population and looking at the population level can allow us to estimate these effects within smaller subsets of the data[61]; Fig. 4E). Indeed, we found that pupil size also predicted changes in how accurately choice information could be decoded from simultaneously recorded populations of FEF neurons. Consistent with our prior results[15], decoded choice probability was significantly lower during exploration compared to exploitation (paired

t-test: both subjects, $p < 0.0001$, Supplementary Fig. 2A). Critically, larger pupil size predicted weaker choice encoding across all trials (Fig. 4F; GLM: beta = -0.0055, $p < 0.0005$, 95% CI = -0.008 to -0.003). This was not driven by differences between the states because pupil size also predicted choice decoding accuracy within exploit trials alone (GLM: beta = -0.003, $p < 0.05$, 95% CI = -0.006 to -0.001). There was no significant effect of pupil size on choice decoding within explore trials (GLM: $\beta = -0.0009$, $p = 0.83$, 95% CI = -0.01 to 0.008, Supplementary Fig. 2B), though decoding accuracy was already close to chance in these trials. These findings show that pupil-linked arousal predicts the strength of choice-predictive neural signals in FEF above and beyond what can be explained by differences between the states.

In our previous work, we found that decreases in choice-predictive activity were accompanied by increases in variability in FEF population responses to the same choice[15]. We quantified this with the "scatter index": a measure of the spread within clusters of same-choice population activity (see Methods). A high scatter index indicates that neural activity on a given trial was dissimilar to other trials where the same choice was made, whereas a low scatter index indicates that neural activity was tightly clustered. We observed a higher scatter index during exploration compared to exploitation (paired t-test: both subjects, $p < 0.0001$, Supplementary Fig. 2C). Here, we also found that increasing pupil size predicted an increase in the scatter index (Fig. 4G; GLM: beta = 0.006, $p < 0.0001$, 95% CI = 0.004–0.009). This effect remained significant and of similar magnitude within exploit trials alone (GLM: $\beta = 0.004$, $p < 0.005$, 95% CI = 0.002–0.007), again suggesting that the relationship between pupil size and scatter was not an artifact of state differences with pupil size. Pupil size again did not significantly predict the scatter index during explore trials (GLM: $\beta = -0.0003$, $p = 0.93$, 95% CI = -0.009–0.0008, Supplementary Fig. 2D). Thus, pupil size predicted disorganization of choice-predictive signals in the FEF, at both the level of single neurons and in the population.

### Neural disorganization mediates the relationship between pupil size and exploration

To test whether neural population activity, like pupil size, also specifically predicted the onset of exploration, we compared its dynamics in the trials preceding exploration to those in matched-reward control trials. While sudden changes in the decoded choice probability and scatter index were largely aligned with the onset of exploration (as reported previously), these neural measures differed on the trials preceding exploration, compared to reward-matched controls (Fig. 4H, I; choice probability, offset = -0.611, $p < 0.001$, $n = 28$; scatter index = 0.258, $p < 0.001$). Reward information did not cause a change in either variable (choice probability, slope = -0.004, $p > 0.5$; scatter index = 0.002, $p > 0.5$). Instead, a small, but significant interaction terms suggested that both variables anticipated the onset of exploration (choice probability interaction = -0.058, $p < 0.01$; scatter index interaction: beta = 0.040, p < 0.001). To determine if these neural measures might explain or mediate some of the relationship between pupil size and exploration, we turned to structural equation modeling[62,63]. We found that the scatter index was a significant mediator of the relationship between pupil size and the onset of exploration (Fig. 4J; effect of mediation, ab = 0.003, $p < 0.005$; full report in Supplementary Table 1). Together, these results suggest that pupil size predicts disruptions in the organization of prefrontal neural activity that then mediate its relationship with the onset of exploration.

### Exploration may reflect a critical transition in brain state dynamics

Neural systems, like other complex networks, can undergo tipping points—irreversible "critical transitions" between stable operating regimes[64–67]. Because exploration occurs as the brain passes from exploiting one target to exploiting another, it is worth considering the possibility that exploration may represent a critical transition in brain states. Indeed, during exploration, we previously reported[15] several phenomena in the FEF and in behavior that are hallmarks of critical transitions, including a rapid flickering back and forth between choices[67], an increase in the variance in neural activity[66], and a disruption of long-term neuronal autocorrelations that suggests that passing through exploration causes time-irreversible changes in the FEF network[65]. However, there is another classic feature of critical transitions that we did not consider: an early warning signal known as "critical slowing". As the system nears the tipping point, the dynamics within the system begin to flatten out in preparation for the change. As a result, the systems' processes slow down and take longer to trace the same paths[66]. Therefore, we next asked if there was any evidence that decision-making slowed down in advance exploration in this dataset.

To test for critical slowing, we examined two measures of decision speed: one behavioral and one neural. First, we looked at response time, a

measure of how long it takes the brain to generate saccadic decisions. Response time was not only slower in the trials before exploration, compared to matched-reward control trials (Fig. 5A–C; GLM offset = 0.39, $p < 0.0001$, n = 28), but it slowed down over trials before the onset of exploration (interaction = 0.05, $p < 0.001$). Second, we looked at the mean rate of change in neural population choice signals during the decision process ("neural speed", see Methods). Neural speed was only weakly correlated with response time across sessions (mean = -0.07, min = -0.36, max = 0.09, Pearson's correlation), suggesting that the measures were complementary, rather than redundant. Like response time, neural speed was also significantly slower on average in the trials before exploration, compared to matched-reward controls (Fig. 5D–F; GLM offset = -0.17, $p < 0.0001$, $n = 28$). However, unlike response time, neural speed did not show a significant interaction (beta = -0.01, $p = 0.08$). Although the notion that the brain may be subject to critical tipping points is controversial[64], these results are consistent with the idea that exploration could reflect a critical transition between exploiting one option and exploiting another.

We first asked whether the typical reward histories that precede exploration were a sufficient explanation for the relationship between slowing and the onset of exploration. However, reward history alone did not have a significant effect on either neural or behavioral slowing (response time: slope of matched-reward trials = 0.0002, $p > 0.5$; neural speed: slope = -0.018, $p > 0.1$). This suggests that some internal variable, like arousal, could be driving increased slowing and, perhaps, also the systems' proximity to a tipping point. Indeed, increasing pupil size predicted slower response times (Fig. 5B; GLM beta = 0.08, $p < 0.0001$, $n = 28$ sessions), even within periods of exploitation (beta = 0.05, $p < 0.0001$). The same was true of neural slowing (Fig. 5E; all trials: beta = -0.03, $p < 0.0005$; exploit only: beta = -0.09, $p < 0.0001$). Further, structural equation modeling revealed that both measures of slowing mediated the relationship between pupil size and the onset of exploration (Fig. 5C, F; Supplementary Table 2–3). In sum, the pupil-linked mechanisms that anticipated exploration included both a disorganization of neural activity and a slowing of decision-related computations in brain and behavior—hallmarks of a system approaching a critical transition.

## Discussion

Random decision-making is a powerful strategy for exploration[9–11,14,15] that is linked to disorganized patterns of neural activity in the prefrontal cortex[10,15,17]. Here, we sought to identify some of the neurobiological mechanisms that drive random exploration and its neural signatures. We found that pupil size, a peripheral correlate of autonomic arousal, predicted exploration and certain measures of neural population activity previously linked to exploration. Consistent with previous studies[36], pupil size was generally larger during exploration, compared to exploitation. However, there was also a complex temporal relationship, where pupil size ramped up between periods of exploration and decreased during exploration. As a result, pupil size was largest at the beginning or "onset" of exploration and explained variance in the onset of exploration that could not be explained by other variables. Together, these results suggest that pupil-linked mechanisms may play a role in driving the brain into an exploratory state.

Our behavioral results largely replicate previous findings linking exploration to increased pupil size[36]. However, where we found gradual ramping before exploration and sudden constriction after, Jepma and Nieuwenhuis (2011)[36] reported an abrupt (if modest) increase of pupil size at the onset of exploration and then a gradual decrease at the return to exploitation. The discrepancy may be due to differences in the operational definition of exploration. Jepma and Nieuwenhuis (2011) fit an RL model to behavior and defined "explore choices" as the choices that were not reward-maximizing according to the model. This definition conflates exploration with errors of reward maximization. A strategy that is non-reward-maximizing would produce choices that are orthogonal to value, not consistently bad. Here, we used an HMM to identify latent explore and exploit states on the basis of the temporal profiles of choices alone, with no assumptions on the underlying value computations. This allowed us to

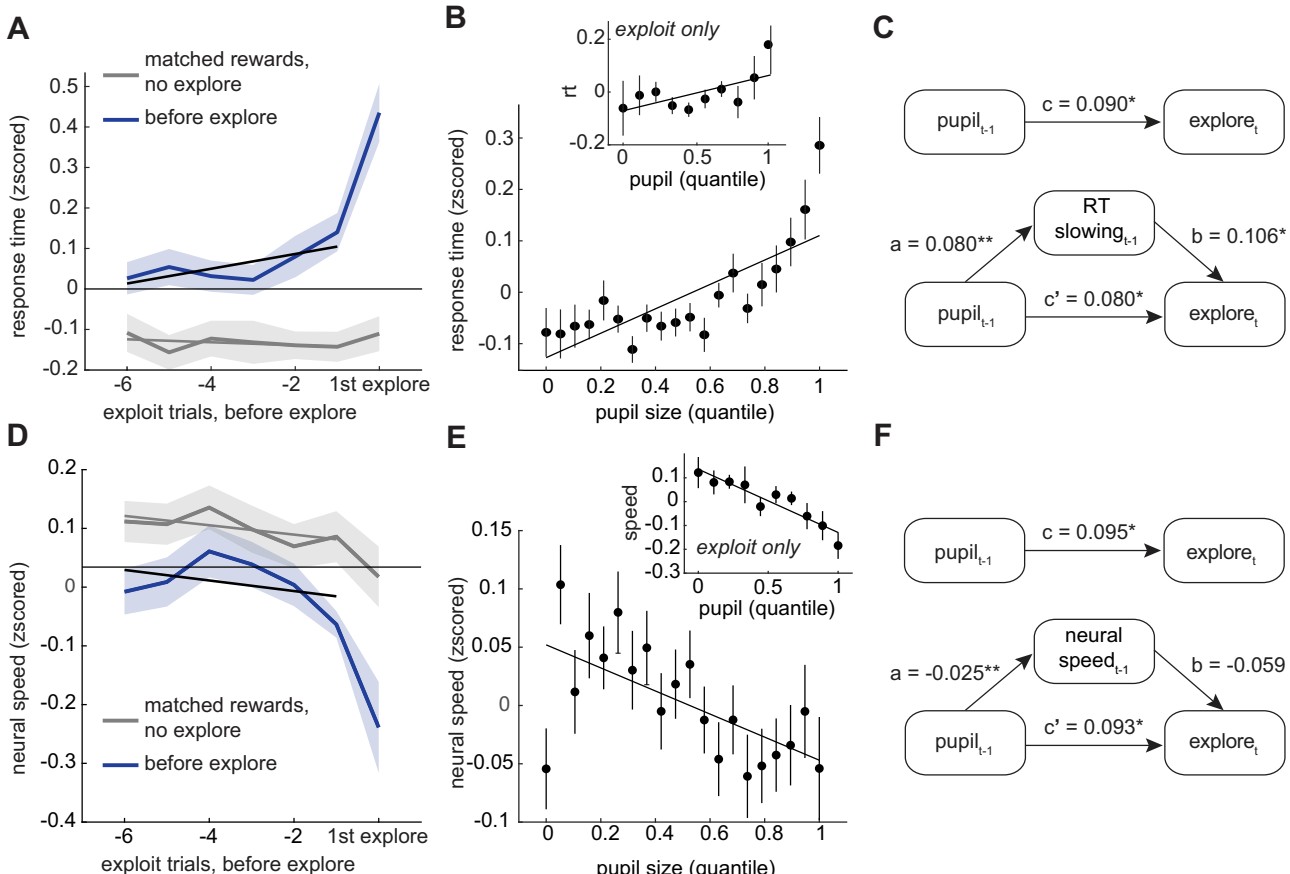

**Fig. 5 | Pupil size predicts behavioral and neural slowing. A** Response time on exploit trials before the onset of exploration (blue) and trials with matched rewards but no exploration (gray). **B** Response time plotted as a function of pupil size quantile. Inset: Same for exploit trials alone. **C** Mediation analysis between pupil size, response time, and the onset of exploration. Top: Direct model. Asterisks marked significant paths (*p < 0.01). Bottom: Indirect, mediated model. **D** Neural speed on exploit trials before the onset of exploration (in blue) and trials with matched rewards (in gray). **E,F** Same as (**B,C**) for neural speed. Error bars and shaded areas depict ± SEM throughout.

dissociate the effects of reward history from the explore/exploit choice labels. We reported here (Fig. 1C), and in previous studies[15,46], that HMM labels outperform RL labels in explaining behavioral and neural measures, suggesting that the HMM may more accurately separate distinct neural and behavioral states. If the HMM allows for more precise identification of exploratory and exploitative choices, it would follow that it also allows for more precise reconstruction of the temporal relationship between the pupil and exploration.

The precision of our explore/exploit labels revealed that the U-shaped relationship between pupil size and exploration was caused by a refractory constriction in the pupil. When exploration was plotted as a function of pupil size, the relationship appeared non-linear: both small- and large-pupil choices were more likely to be exploratory. This superficially resonated with the idea of a U-shaped relationship between arousal and task performance (i.e., the "Yerkes-Dodson curve"[31,68]: perhaps reliable exploitation is only possible at intermediate levels of arousal). However, when we examined the temporal relationship between exploration and pupil size, we found that pupil size only predicted the onset of exploration, the first explore choice in a sequence. Small-pupil explore choices happened because starting to explore seemed to "reset" the level of pupil-linked arousal, causing it to quickly fall below baseline. If increased pupil size promotes a transition to exploration, then it is possible that post-exploration constriction represents a refractory period for exploration. Given that uncertainty grows with time in this task (and in all dynamic environments), it may not be smart to start to explore again immediately after you have just explored. A refractory period could ensure that non-reward-maximizing explore choices are deployed only

when needed. Future work is needed to test this hypothesis and to determine the cognitive and/or neurobiological mechanisms at play.

Before exploration, we observed an oscillatory dynamic that was about twice as fast as the 15 trials it took the pupil to recover after exploration. This 0.06-0.09 Hz oscillation entrained the onset of exploration: onsets tended to occur during the rising phase of pupil size, whereas identical trial sequences that did not result in exploration were on the opposite phase. This implies that it is the confluence of pupil size, pupil phase, and trial history that best predicts the onset of exploration. This result reinforces the idea that arousal or arousal-linked mechanisms help trigger random exploration[19,25,27], rather than just tracking the reward-linked variables that make exploration more probable. It is also notable that the period of the pupil oscillation was close to the frequency of the Mayer wave: an oscillation in blood pressure that entrains other autonomic measures, including respiration and heart rate[51–55]. There is precedent for the idea that behavior can be entrained by the Mayer wave: in marmosets, fluctuations in arousal predict the spontaneous onset of a call[51]. This paper argued that the Mayer wave may function to organize vocal communication by bringing the system closer to the threshold for transitioning from inaction to action. It is possible that oscillations in the pupil and pupil-linked mechanisms function the same way here, organizing important state changes in time. In parallel, pupil-linked mechanisms seem to anticipate other state transitions, including belief updating[35,39], task disengagement[69], and other behavioral state changes[32]. Together, these results suggest an important role for pupil-linked mechanisms in driving successful transitions between certain neural and behavioral states.

Critically, pupil size and pupil oscillations did not predict all state transitions here, but only the transition into exploration. What kinds of state transitions might be entrained by pupil- linked arousal? It is possible that the pupil may have a special relationship with certain "critical" kinds of transitions. Critical transitions are abrupt, large-scale, and irreversible changes in the dynamics and behavior of complex systems, like the brain. As these systems go from being in one conformation (i.e. always choosing the left option) into another conformation (i.e. always choosing the right), the system dynamics that support the old state have to disappear and the new dynamics have to emerge. During this brief transitory period, when both dynamics co-exist in the system, certain signatures can be observed in the system. We previously reported that the exploration was accompanied by abrupt changes in neural population activity, certain patterns of noise in brain and behavior, and disruptions in long-term neuronal autocorrelations: all observations that could be interpreted as suggesting that exploration is a critical transition in the brain[15]. Here, we found that pupil size predicts these features of neural activity and also an prominent "early warning sign" of critical transitions: a slowing, in brain and behavior, of the decision process. While there are certain patterns of activity in FEF that predict response speed[70,71], here we identified *independent* neural and behavioral measures of decision speed that both mediated the relationship between pupil size and exploration. Notably, pupil size also predicted slower neural and behavioral responses within exploit-only trials. This suggests that these effects are not an artifact of differences between explore and exploit states, but this result also makes it impossible to interpret these patterns of neural activity as specific signatures of the onset of exploration itself. Instead, it is more likely that arousal has a domain-general influence on neural dynamics, causing ongoing fluctuations in information encoding and speed. In this view, the transition into exploration in FEF may best be thought of as the result of a critical tipping point that occurs when the conditions necessary for exploration to occur (i.e., domain-general fluctuations neural dynamics linked to arousal) happen to coincide with evidence in favor of exploration (i.e., sequences of omitted rewards). Together these results highlight the crucial role of pupil-linked mechanisms in changing the dynamics of the brain.

What underlying, pupil-linked mechanisms could support critical transitions? Changes in pupil diameter coincide with neuromodulator system activity, especially norepinephrine (NE) and acetylcholine[27,29,72–75]. At the neuronal level, central NE flattens tuning curves, at least in the auditory cortex[30], though it may have different effects in non-cortical structures[76]. Here, we made a parallel observation: as pupil size increases, neuronal turning curves flattened, and choice-predictive neural population activity became disorganized. These results resonate with a particularly influential theory of NE function: the idea that NE release may facilitate "resets" in cortical networks in order to effect long-lasting changes in brain and behavior[31,32]. More recent studies seem to consistently report that elevated levels of NE predict an increase in behavioral variability, while pharmacological blockade of NE receptors reduces variability[24,69,77,78]. In combination with the present study, these results could suggest that phasic NE signaling functions to push the brain towards a critical tipping point where it is better able to transition from one regime to another. In this view, behavioral variability would be linked to NE not because NE increases variability directly, but because the brain is more likely to transition into a high variability regime after it is released. Of course, pupil size is also associated with other neuromodulatory systems, cognitive factors, and other measures of arousal. It also remains unclear whether similar results would be obtained in other species and tasks. Thus, future work is needed to identify the neurobiological mechanisms that underpin the relationship between pupil size and critical transitions that we report here.

## Materials and Methods
### Surgical and electrophysiological procedures
All procedures were approved by the Stanford University Institutional Animal Care and Use Committee. We have complied with all relevant ethical regulations for animal use. Subjects were two male rhesus macaques (aged 5–10 years), surgically-prepared with head restraint prostheses, craniotomies, and recording chambers under isoflurane anesthesia. Following surgery, analgesics were used to minimize discomfort, and antibiotics were delivered prophylactically. After recovery, subjects were acclimated to the laboratory and head restraint, then placed on controlled access to fluids and trained to perform the task. Animals were monitored daily by veterinary staff; no adverse events were observed. Experiments would have been stopped immediately if any animal had shown signs of distress or infection, in accordance with veterinary guidance and institutional policies.

In order to train the animals on the explore/exploit task a gradual procedure was used in which the two animals were first trained to make saccadic eye movements in exchange for liquid rewards. Once the animals reliably made controlled eye movements to a single target (generally within 1–2 days), a second target was introduced, and the animals were free to choose between them. At the outset, each target was associated with a probability of reward (initially 10% and 90%), which was reversed in blocks at the experimenter's discretion. Over a period of 2–4 months, the difference in reward probabilities between the targets was gradually reduced, the blocks transitioned into gradual reward probability shifts (reward walks), and a third target was introduced. The speed and order of these changes depended on each animal's performance and engagement with the task. One animal (monkey O) was naïve to laboratory tasks prior to this experiment, whereas the second (monkey B) had been previously trained on covert and overt attention tasks, but not on any prior value-based tasks.

Recording sites were located within the FEF, which was identified via a combination of anatomical and functional criteria. The location of recording sites in the anterior bank of the arcuate sulcus was verified histologically in one subject and via microstimulation in both subjects[15]. Recordings were conducted with 16-channel U-probes (Plexon), located such that each contact was within gray matter at an FEF site. An average of 20 units were recorded in each session (131 single units, 443 multi units; 576 total units across 28 sessions).

### General behavioral procedures
Eye position and pupil size were monitored at 1000 Hz via an infrared eye tracking system (SR Research; Eyelink). The manufacturer's standard methods for calculating pupil area were used. MATLAB (Psychtoolbox-3[79]) was used to display stimuli and record behavioral responses and pupil size measurements. Task stimuli were presented against a dark gray background (7 cd/m2) on a 47.5 cm wide LCD monitor (Samsung; 120 Hz refresh rate, 1680 × 1050 resolution), located 34 cm in front of the subject.

### Three-armed bandit task
The subjects performed a sequential decision-making task in which they chose between three targets whose values changed over time. The subject first fixated a central fixation square (0.5° stimulus, ± 1.5-2° of error) for a variable interval (450–750 ms). At any point within 2 s after the onset of the targets, subjects indicated their choice by making a saccade to one of the targets and fixating it (± 3°) for 150 ms. Reward magnitude was fixed within session (0.2-0.4 μL). Reward probability was determined by the current reward probability of the chosen target, which changed independently over trials for each of the three targets. On every correct trial, each target had a 10% chance of the reward parameter changing either up or down by a fixed step of 0.1, bounded at 0.1 and 0.9. Because rewards were variable, independent, and probabilistic, the subjects could only infer the values of the different targets by sampling them and integrating noisy experienced rewards over multiple trials.

### General analysis procedures
Primary measures included choice behavior (exploit vs. explore), pupil diameter, saccade peak velocity and reaction time, and various measures of neural population activity in the frontal eye field brain region. Data were analyzed using custom software written in MATLAB. The values of behavioral and neural variables were z-scored within each session to facilitate comparisons across sessions; in the results, we refer to a z-score of 0 as "baseline." For all choice-predictive neural measures, firing rates were averaged over a 200 ms window immediately preceding target onset.

A longer, whole-trial epoch was chosen for neural speed analyses (0 to 400 ms following target presentation). For analyses of behavioral or neural variables on the trials immediately preceding or following exploration, we required continuous runs of exploit trials. Model comparison was based on information-theoretic methods: we computed the likelihood of the data and the Akaike information criterion (AIC) for each model, then used AIC weights to identify (1) the model most likely to minimize information loss and (2) the relative support for competing models (Burnham and Anderson, 2004).

## Statistics and reproducibility

All statistical analyses were performed in MATLAB. Sample size was based on standard practice in neurophysiological studies to ensure stable recordings and adequate trial counts for our primary outcomes (changes in pupil size and prefrontal neural activity around the onset of exploration). No formal a priori power calculation was performed. Unless otherwise noted, statistical tests were paired, two-sided $t$-tests, and generalized linear models were fit to raw data, with session number included as a dummy-coded factor to account for session-to-session variability. Normality was assessed by visual inspection and nonparametric alternatives were applied when data was non-normal (e.g., Wilcoxon rank sum). Behavioral and neural analyses were based on data from two male rhesus macaques, across 28 recording sessions and 21,793 valid trials. Because recordings were acute, each session provided an independent replication (monkey O: 10 sessions; monkey B: 18 sessions), and all major results were verified in both animals. Monkey O contributed 8396 valid trials (1008 explore trials, 7388 exploit trials), and monkey B contributed 13,397 valid trials (2273 explore trials, 11,124 exploit trials). All analyses were performed on these explore and exploit trials within each animal.

## Pupil size

Pupil size was measured during the first 200 ms of fixation, a time at which the eye was fixed at a known point on the screen, illumination was identical across trials, and anticipatory changes in the pupil were minimal. To remove any blinks or movement artifacts, trials where pupil size or the change in pupil size from the first-time bin of this epoch to the last was ± 6 standard deviations from average were eliminated from further analyses. A total of 178 trials (out of 21,793, approximately 0.8% of observations) were outliers. No sessions or animals were excluded. Criteria were defined a priori and applied uniformly across sessions.

## Hidden Markov Model

To identify when subjects were exploring versus exploiting, we employed a hidden Markov model[15,46]. In this framework, choices are treated as emissions from a latent decision-making state, which can either be an explore or one of the multiple exploit states.

The emission model for exploration assumed a uniform probability of selecting any option:

$$p(y_t = k | z_t = explore) = \frac{1}{N_k}$$

where is the total number of options. In contrast, exploit states deterministically emitted choices to the exploited option $i$:

$$p(y_t = k | z_t = exploit_i, k = i) = 1$$

$$p(y_t = k | z_t = exploit_i, k \neq i) = 0$$

Latent state transitions followed a Markov process, such that the probability of the current state depended only on the previous state:

$$p(z_t | z_{t-1}, y_{t-1} \ldots, z_1, y_1) = p(z_t | z_{t-1})$$

To reduce model complexity, parameters were shared across exploit states, and subjects were assumed to begin in the explore state. The final HMM included only two free parameters: the probability of persisting in exploration and the probability of persisting in exploitation. The model was fit using expectation-maximization with 20 random restarts, and the solution maximizing the observed data log-likelihood was selected. The most probable sequence of latent states was recovered using the Viterbi algorithm.

## Reinforcement learning model

To compare goal state labels derived from an RL and HMM model, we employed a Rescorla-Wagner model. This was fit using maximum likelihood estimation. The value of each option is iteratively updated according to:

$$V_{i,t+1} = V_{i,t} + \alpha(r_t - V_{i,t})$$

Where $V_{i,t}$ is the value of option $i$ at time $t$, $r_t$ is the reward at time $t$, and $\alpha$ represents the fitted learning rate, which determines how much the difference between the predicted and actual reward (the prediction error) influences value. To make a decision, the values are passed through a softmax decision rule:

$$p_{i,t} = \frac{\exp(\beta V_{i,t})}{\sum_{j=1:n} \exp(\beta V_{j,t})}$$

Where $n$ is the total number of available options, and $\beta$ is the inverse temperature, which controls the level of random noise in decision-making. Following previous work[7,8,36], choices that did not maximize expected reward were classified as exploratory (i.e. any decision where $V_{chosen,t}$ was not the maximum V at time $t$).

## Generalized Linear Model (GLM)

To examine the relationship between behavioral and neural variables, we employed generalized linear models (GLMs). These models were fit using maximum likelihood estimation. The dependent variable $Y_i$ (i.e., scatter index, response time, or choice probability) was modeled as a linear combination of predictors:

$$Y_i = \beta_0 + \beta_1.explore\ state_i + \beta_2.pupil\ size_i + \beta_3.(explore\ state_i \times pupil\ size_i) + \varepsilon_i$$

where $Y_i$ is the dependent variable on trial $i$, $\beta_0$ is the intercept, and $\beta_1$, $\beta_2$, $\beta_3$ are regression coefficients quantifying the influence of the corresponding predictors (e.g., explore state, pupil size, and their interactions). $\varepsilon_i$ is the residual error term.

In analyses where categorical variables (e.g., explore state: 0 = exploit, 1 = explore) were used as predictors, these were coded as binary dummy variables. Models were fit using the identity link function and assumed normally distributed errors.

## Learning Index

To investigate whether learning differed with pupil size within the exploratory choices, we calculated a learning index that captured the effect of rewards experienced during exploration on future choices. Because reward effects decay exponentially quickly (Lau and Glimcher, 2008), a 1-trial-ahead index should capture most of the variability in how much is learned between trial types. The equation was:

$$learning\ index_t = \frac{p(switch_{t+1} | reward_t) - p(switch_{t+1} | no\ reward_t)}{p(switch_{t+1})}$$

## Lagged change in pupil size

To determine whether exploration impacts pupil size, we measured the change in pupil size (Δ pupil) between pairs of trials that either were or were not separated by at least one explore trial. Segments of twenty-five

consecutive trials were identified that either included a single bout of exploration or did not include exploration. For each pair of trials within these sequences, we then measured the change in pupil size between the first exploit trial of the sequence (t1) and the remaining exploit trials in the sequence (t2:25). This was repeated for all unique pairs of trials that met our selection criteria.

## Matched reward trials

To test whether the rising trend in pupil size before exploration is best explained by reward history, we identified trial sequences with identical reward and state histories that did not end in exploration ("matched rewards"). For each onset of exploration preceded by at least six exploit trials, we searched for identical sequences of exploit trials, with identical reward histories, that did not end in exploration. We chose six previous trials because this was the longest sequence of reward history we could regularly match within the majority of sessions (we could find at least 10 matched sequences in 96% [27/28] of sessions for six trials sequences; that dropped to 75% [21/28] at 7 trials). Identical results were obtained with other sequence lengths, though these analyses included fewer sessions.

## Mediation analysis

To determine if the predictive relationship between pupil size and exploration was mediated by other variables, we used structural equation modeling to test for mediation. Mediation analyses involve fitting three regression models. The first model measures the total effect (c) of the independent variable (here, pupil size) on the independent variable (here, onset of exploration):

$$explore_t = \gamma_1 + c(pupil_{t-1}) + \epsilon_1$$

In these equations, $\gamma$ represents the intercept for each equation, while $\epsilon$ represents the error of the model. Note that the estimated parameter c will include both direct effects of pupil size on exploration, but also indirect effects that may be mediated by other variables. Therefore, we also fit a second model that tests if the independent variable also predicts a potential mediator variable (here, neural network scatter):

$$scatter_{t-1} = \gamma_2 + a(pupil_{t-1}) + \epsilon_2$$

Model parameter a thus captures the effect of pupil size on the mediator. Finally, a third model estimates the unique contributions of both the potential mediator (scatter, b) and the independent variable (pupil size, c'), now controlling for the mediator:

$$explore_t = \gamma_3 + b(scatter_{t-1}) + c\prime(pupil_{t-1}) + \epsilon_3$$

A drop between c and c' indicates that the effect of the independent variable (pupil) on the dependent variable (exploration) is reduced when the mediating variable is considered. The mediation effect (the indirect effect of the pupil size on the onset of exploration via the mediating factor) can also be estimated directly, via taking the product of the coefficients a and b. Sobel's test is used to determine the significance of the mediation path[63].

## Phase analysis

To determine if the onset of exploration happened at a specific phase of pupil size over trials, we performed a wavelet analysis. Because this method only assumes local stationarity, it is more suitable than other methods for analyzing pupil size, which tended to ramp over trials. A wavelet was constructed by multiplying a complex sine wave (frequency = 5 trials) with a Gaussian envelope ($\mu = 0$, $\sigma$ = cycles / ($2\pi$*frequency), cycles = 5[80];). The wavelet was convolved with the baseline pupil size time series and the phase of the signal was calculated on each trial (Matlab; angle). Standard circular statistics were used to measure the differences between phase distributions for explore onsets and reward-matched controls[81] and the phase alignment

within these trial types[82]. The latter was also verified via comparison with bootstrapped null distributions (1000 samples).

## Targeted dimensionality reduction

Neural state spaces have as many dimensions as there are recorded neurons, but converging evidence suggests (1) that the neural states that are observed in practice are generally confined to a lower-dimensional "manifold", and (2) that task-relevant information is encoded by a small number of dimensions in that manifold. Because we wanted to isolate the effects of arousal on choice-related activity from well-known effects of arousal on neural activity[21–23,27,83–86], we focused all our neural population analyses on activity within the choice-predictive subspace, rather than on neural activity more broadly.

To do this, we used targeted dimensionality reduction to identify the choice-predictive dimensions of the neural state space[15,87–89]. Specifically, we used multinomial logistic regression (Matlab; mnrfit, mnrval[90],) to identify the separating hyperplanes that best discriminated each choice from the alternative choices. This is equivalent to fitting a system of binary classifiers of the form:

$$p(choice = k \mid X) = \frac{1}{1 - \exp^{-\beta_k X}}$$

Where one classifier discriminates target 1 choices from targets 2 and 3 and a second discriminates target 2 choices from targets 1 and 3. The classifier that discriminates target 3 from targets 1 and 2 is then just the negative of target 1 and target 2. These axes span the subspace in which neural activity best predicts choice. Classifiers were trained on firing rates from an epoch that began when the targets appeared and ended at the time of the saccade. Mean imputation was used for the small number of occasions where a unit was not held for the whole duration of the session ( ~ 3% of trials, ~12% of units) and a small fraction of units were omitted from these analyses because their mean firing rates were less than 2 spikes/s, which makes their weights difficult to identify ( ~ 8% of units).

## Choice Probability Decoding

Within the choice-predictive subspace, the distance from the separating hyperplanes (the vectors illustrated in Fig. 4E) are the decoding vectors: the vectors along which we can project neural activity in order to decode the log odds of choice. This projection is equivalent to the decoded choice probability from the multinomial logistic regression model and this is the figure we took as the decoded choice probability in Fig. 4F, H. We evaluated decoding accuracy by measuring how often the most-likely choice predicted by the model coincided with the choice the subject made.

## Scatter index

The scatter index measures how much choice-predictive population neural activity is clustered between trials with the same choice[15]. It is calculated by measuring the average Euclidean distance of each trial from all other trials where the same choice was made and dividing it by the average Euclidean distance to all other trials where a different choice was made:

$$scatter = \frac{d_{within}}{d_{between}}$$

Each trial thus has its own scatter index value, with a value of 1 indicating no difference in clustering between same-choice and different-choice trials, and a value less than 1 indicating greater clustering with same-choice trials compared to different-choice trials.

## Neural speed

To determine how the speed of the decision-making process changed before and during exploration, we calculated the rate of change in neural states within the choice-predictive subspace during the first 400 ms following

target presentation. Each trial's neural activity was sampled in non-overlapping 20 ms bins and then projected into the choice-predictive subspace. The change in neural activity within the subspace was then calculated between each pair of samples. Finally, the changes were averaged together across the trial and normalized to the bin width to produce an average rate of change in choice-predictive activity for that trial.

## Reporting summary

Further information on research design is available in the Nature Portfolio Reporting Summary linked to this article.

## Data availability

The datasets generated during and/or analyzed during the current study are available in the Zenodo repository, https://doi.org/10.5281/zenodo.17546524 (ref. 91).

## Code availability

Custom MATLAB scripts used to produce the figures for this study are available from the corresponding author upon request.

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

## Acknowledgements

This work was supported by the Natural Sciences and Engineering Research Council of Canada (Discovery Grant RGPIN-2020-05577), the Fonds de Recherche du Québec–Santé (Junior 1 Chercheur-Boursier 284309 to R.B.E.), the Jacobs Foundation (Research Fellowship, seed grant to R.B.E.), and CIFAR Azrieli Global Scholars (seed grant to R.B.E.), the National Eye Institute (R01-EY014924 to T.M.), and l'Institut de valorization des données (fellowship to K.J.).

## Author contributions

Conceptualization: B.E. and T.M. Methodology: A.S., M.A., and B.E. Formal analysis: A.S., G.M., K.J., and M.A. Investigation: A.S., M.A., G.M., and B.E. Data curation: B.E. Writing – original draft: A.S., M.A., G.M., and B.E. Writing – review & editing: A.S., M.A., and B.E. Visualization: A.S., M.A., G.M., K.J., and B.E. Supervision: B.E. and T.M. Funding acquisition: T.M. and B.E.

## Competing interests

The authors declare no competing interests.
