## [Transparent Peer Review file · Communications Biology]

Pupil size predicts exploration through critical slowing in prefrontal dynamics

Corresponding Author: Dr R. Becket Ebitz

Version 0:

Reviewer comments:

Reviewer #1

(Remarks to the Author)

The researchers measured pupil size and recorded neural activity from prefrontal neurons in two rhesus macaques as they performed a decision-making task. They found that the pupils were consistently larger during exploratory choices than during exploitative ones. Based on these data, they suggest that pupil-related mechanisms drive prefrontal cortex activity toward a critical tipping point, encouraging exploratory behavior. This manuscript is logically structured and presents a convincing argument, but some concerns remain, as outlined below.

The meaning of the title is ambiguous, making it difficult to understand the content. What exactly does 'in brain and behaviour' refer to? A more specific title that better reflects the content of the paper is desirable.

In Figure 1F, several behavioral parameters are analyzed in relation to pupil size. However, reaction time is known to be an important factor in analyzing choice behavior. This should at least be mentioned in the text.

Figure 3D does not use actual data but rather serves as a validation of a hypothetical model derived from the experimental results. To avoid confusion for readers, this content should be included in the Discussion section instead."

Reviewer #2

(Remarks to the Author)

In this study, the authors used a three-armed bandit task to examine the role of pupil dilation in predicting transitions between exploratory and exploitative internal states, as estimated by a Hidden Markov Model. They further investigated how pupil activity influenced frontal eye field (FEF) neuron activity driving choice-related saccades. The authors found that exploration states were associated with larger pupil size compared to exploitation states, and that the relationship between pupil size and the probability of exploring followed a U-shaped function. Pupil size ramped up across trials preceding the onset of an exploration state and began shrinking immediately afterward, suggesting that pupil dilation plays a role in initiating—but not maintaining—exploration. This dynamic may explain its U-shaped relationship with exploration probability. Control analyses indicated that reward history might contribute to this ramping effect, though an offset remained specific to trials preceding exploration states. Further analyses suggested that the link between larger pupil size and exploration states could be driven by both reward history and arousal fluctuations over time. At the neural level, larger pupil size was associated with increased disorganization in FEF neuronal choice encoding, and this disorganization mediated the pupil's role in triggering exploration. Additionally, pupil size influenced both neural speed and response times, suggesting a role in decision-making slowing at both behavioral and neural levels. Overall, the study is well-structured, addresses important questions, and provides novel insights into the relationship between pupil size, neural activity, and exploration. However, I have some concerns regarding the organization of the results and the clarity of certain aspects.

MAJOR COMMENTS

Results

• I find somewhat difficult to follow the narrative while reading the Results section. While the logical progression aligns with the analysis workflow, it makes it challenging for the reader to grasp and integrate key takeaways throughout. For example,

by the time the reader reaches line 302, the initial conclusion—that pupil ramping directly predicts the onset of exploration—is reconsidered in light of control analyses suggesting a role for time and reward history. This leads to a shift in focus toward further analyses. While the control analyses are highly valuable, they seem to diverge from the central message, making the narrative feel overwhelming and somewhat distracting.

Consider streamlining the Results section to maintain a clearer emphasis on the main findings while still incorporating the necessary control analyses in a way that enhances, rather than diffuses, the key message.

- [Lines 184-202] The rationale behind the median split analyses needs clarification. Specifically, it would be helpful to explain why the authors selected reward, peak velocity, scatter index, and learning index for their control analyses and how targeting these dimensions helps exclude disengagement effects.
- In general, authors refer to a “baseline” all along the results section, but it is not clear what they mean by baseline. It seems to correspond to $z\text{-score} = 0$, but this should be explicitly stated to facilitate reading. Additionally, this information is missing from the Methods section and should be included.
- Authors should consider adding labels to the task procedure in Figure 1A, with information about the neuronal recording period as well. Also, clarify the meaning of T1, T2, and T3 in this figure, as this only becomes explicitly clear in Figure 4.
- In Figure 1C It is unclear what type of beta weights are reported. Are these from a GLM, an HMM/RL model, or something else? Please specify in the figure description and/or the Methods section.
- The meaning of the dotted line in Fig 4B is unclear. Please explain it in both the text and the figure legend.
- Figure 4D: The term “All Tuned Neurons” is ambiguous. Clarify whether this refers to 88 out of 155 neurons or 21 out of 88.
- Figures 4F and 4G: The main panels show all trials, while the insets depict exploit-only trials. Given that the study focuses on explore trials, it would be more relevant to show explore-only trials in the insets for both decoding and the scatter index.
- [Line 387] Since the authors report a decrease in decoded choice probability during explore choices compared to exploit choices, it would be informative to include a plot directly comparing these two conditions.
- Figures 4J-5C-F/Tables of Supplementary: There are inconsistencies in reporting the total effect (c). In Figure 5C, effect values (c , a , b , c') are rounded to the second decimal place, while other figures and tables use three decimals. Additionally, the total effect is reported as 0.090 in Figure 4J, 0.09 in Figure 5C, and 0.095 in Figure 5F. If these values are derived from identical analyses, the discrepancy should be explained. If they reflect different analyses, this distinction should be made clear.

Materials and Method

- It is not clear if the animals were trained before the experiment and how. If this was the case, as I assume, I would recommend that the authors introduce training information in the Materials and Method section.

MINOR COMMENTS

- [Line 146] – The acronym “FEF” appears here for the first time. For clarity, spell it out in full before introducing the abbreviation.
- [Line 175-176]: The phrase “there was also a U-shaped relationship between pupil size and the probability of switching” is ambiguous. Clarify whether “switching” refers to a switch choice or a state switch.
- Figure 1F The text refers to a “blue line” [Line 157], but no blue line appears in the figure. Additionally, the panel order in Figure 1F does not match the order described in the text. Consider placing the “learning index” as the last panel to maintain consistency.
- [Lines 224-225] Specify which neural measures are being referred to in this section.
- Figure 4H-I-J There is a mismatch between panel labels and their descriptions in the legend:
 - o Panel H corresponds to the Figure 4I legend.
 - o Panel I corresponds to the Figure 4J legend.
 - o Panel J corresponds to the Figure 4H legend.Please update the figure legend accordingly.
- [Lines 399-400] The phrase “both subjects” in “we observed that increasing pupil size predicted an increase in the scatter

index in both subjects (Figure 4G; GLM: $\beta = 0.04$, $p < 0.0001$)" is unclear. Specify which subjects are being referred to.

- [Line 451-455] – The results for neural speed are different compared to those of response times, but the statements accompanying stats read ambiguous. Specifically, for response times, the authors report that "Response time was not only slower in the trials before exploration [...] but it slowed down over trials before the onset of exploration". For neural speed, they say that "Like response time, neural speed was also significantly slowed in the trials before exploration, compared to matched-reward controls". Since the interaction term is not significant for neural speed, this distinction should be explicitly stated to avoid ambiguity.

- Since no figures are referenced in the Discussion except Figure 1C, consider removing this mention for consistency.

Typos

[Line 448]: missing bold font for Figure 5A-C.

[Line 496]: "latent explore and explore states" should be corrected in "latent explore and exploit states".

[Line 385]: A double closing parenthesis is present, please correct it.

Reviewer #3

(Remarks to the Author)

The authors investigated the role of pupil size in exploration behavior and its underlying neural mechanisms using simultaneous recordings of pupil size and frontal eye field (FEF) activity in an explore/exploit task in two behaving monkeys. They show a U-shaped relationship between tonic pupil size and exploration probability. Moreover, pupil size linearly correlates with the onset of exploration, and this effect remains pronounced even after controlling for reward history. More importantly, pupil size is linked to FEF activity and saccadic reaction times. Overall, the manuscript is very well-written, the experiments are well-designed, and the results are highly compelling. These findings contribute significantly to the literature on exploration-exploitation behavior, demonstrating that pupil-linked mechanisms mediate the onset of exploration by altering prefrontal cortex activity. The main limitation is that "neuronal tuning functions are too noisy to partial out the contributions of exploration and pupil size," preventing detailed analyses of the relationship between pupil size and FEF activity in the context of exploration behavior. Beyond this, I have only a few relatively minor comments.

The authors argue that the U-shaped (instead of linear) relationship observed between pupil size and exploration choice probability (Fig. 1E) is driven by pupil size from non-first exploration trials. While some indirect evidence is provided (e.g., Fig. 2D), the authors do not present direct evidence to support this argument. It is suggested that they plot Fig. 1E to analyze the probability of non-first exploration as a function of pupil size. According to their hypothesis, a smaller pupil size should be associated with a higher exploration probability.

One of the main findings of the manuscript is the linear relationship between pupil size and exploration onset (Fig. 2E). It is therefore important to present individual monkey data to confirm that both monkeys exhibit a similar pattern of results.

"A total of 88 out of 155 single neurons were tuned for choice, and of those, 21 were also modulated by pupil size, while 16 showed a significant interaction between choice and pupil size." This indicates that 43% of FEF neurons are not modulated by choice. This raises an interesting question: Are these 43% of FEF neurons modulated by pupil size? If some portion of these "task-independent" FEF neurons is influenced by pupil size, would the same effects be observed in the relationship between pupil size and neural speed (Fig. 5E)? If so, this may suggest that pupil-linked mechanisms operate in a more domain-general manner, affecting not only exploration-exploitation behavior but other types of behavior as well.

Similar effects were observed in exploit-only trials (Fig. 5B and 5E), suggesting that the link between pupil size and reaction time (or neural speed) may represent a more general effect, independent of the exploration-exploitation context.

Version 1:

Reviewer comments:

Reviewer #2

(Remarks to the Author)

The authors have done a remarkable job at addressing my prior comments and suggestions, and I have no remaining follow-up questions or comments.

Reviewer #3

(Remarks to the Author)

The authors have clearly addressed all of my previously raised concerns. I have no further comments.

Response to Reviewers

We would like to thank the reviewers for their careful review of our manuscript and their thoughtful comments. We have edited the manuscript to address each of the comments, and believe the work has improved as a result.

Below, we provide point-by-point responses to the reviewers' comments. For clarity, in this document, we have included all of the reviewers' comments in black, our responses in blue, and quotes from the revised manuscript in red, with new text in bold. In the manuscript all major changes are highlighted with yellow.

Reviewers' comments:

Reviewer #1 (Remarks to the Author):

The researchers measured pupil size and recorded neural activity from prefrontal neurons in two rhesus macaques as they performed a decision-making task. They found that the pupils were consistently larger during exploratory choices than during exploitative ones. Based on these data, they suggest that pupil-related mechanisms drive prefrontal cortex activity toward a critical tipping point, encouraging exploratory behavior. This manuscript is logically structured and presents a convincing argument, but some concerns remain, as outlined below.

We appreciate the reviewer's summary and comments about the logical and convincing nature of the manuscript and hope the revision addresses their important concerns.

1) The meaning of the title is ambiguous, making it difficult to understand the content. What exactly does 'in brain and behaviour' refer to? A more specific title that better reflects the content of the paper is desirable.

To address the reviewer's concern, we have updated the paper title to better highlight the central result of the paper:

Pupil size predicts exploration through critical slowing in prefrontal dynamics

2) In Figure 1F, several behavioral parameters are analyzed in relation to pupil size. However, reaction time is known to be an important factor in analyzing choice behavior. This should at least be mentioned in the text.

We thank the reviewer for this suggestion, and we have now analyzed reaction time as a function of pupil size. The results are consistent with the other behavioral and neural variables and are now shown in main text Figure 1F. Briefly, the result is consistent with our findings for other control variables in Figure 1F. We also found no significant difference in reaction time between high- and low-pupil explore trials (mean difference = 0.02 ± 0.29 STD, $p > 0.6$, $t(27) = 0.48$), again suggesting that small-pupil explore choices are not associated with disengagement or slowed responding. We have updated the Results text accordingly (page 8) and caption of Figure 1 to include this result.

Result section:

Reaction times were also similar across small- and large-pupil explore choices (mean difference = 0.01 ± 0.02 STD, $p > 0.6$, $t(27) = 1.84$, Cohen's $d = 0.35$; explore/exploit AUC = 0.58 ± 0.06 STD).

Figure caption:

*F) Several behavior measures compared across median-split large- and small-pupil-size explore choices. Left to right: reward probability, a one-trial-back learning index (see Methods), saccadic peak velocity of saccades, the scatter index, **and reaction time**. No significant differences between pupil bins.*

3) Figure 3D does not use actual data but rather serves as a validation of a hypothetical model derived from the experimental results. To avoid confusion for readers, this content should be included in the Discussion section instead.

The original presentation order was confusing. The hypothesis illustrated in Figure 3D was not intended as a discussion point, but rather to illustrate the hypothesis motivating the analyses shown in the original Figure 3B and 3C. Because the concept of pupillary phase is critical for the reader to understand these two analyses, we have reordered the figure panels: the original panel D is now presented as the new panel B, clearly marked as a hypothesis with a dashed-line frame, and the original panels B and C are now presented as panels C and D to illustrate the testing of this hypothesis. We believe this approach balances the explanatory value of the schematic with the need to make its illustrative nature clear to the reader.

We have also updated the Results section on pages 13–14 to reflect these changes and to reduce the potential for confusion.

Visual inspection of Figure 3A suggested that there may be a phase difference in pupil size between trials where exploration began and matched reward trials where exploration did not begin. This led us to develop a novel hypothesis (Figure 3B): that rewards may interact with ongoing oscillations in pupil size. Due to delays in communication between the baroreceptor reflect and changes in heart rate, the sympathetic nervous system (Borjon et al., 2016; Japundzic et al., 1990; Julien, 2020, 2006; Kamiya et al., 2005; Liao et al., 2018) has a natural oscillation known as the Mayer wave, with a period of approximately 0.05–0.1 Hz (Borjon et al., 2016b; Julien, 2006). Critically, transitions in other behavioral states can be entrained by this oscillation, with eliciting stimuli causing transitions only at certain phases of arousal. If pupil size is out of phase between trials that lead to exploration and trials that do not, then the previous trials most predictive of the onset of exploration may actually be several trials prior to onset itself—during the periods in which out-of-phase signals would be most distinct. Indeed, the trials in which pupil size best predicted the onset of exploration were not those immediately preceding it (e.g., trial $t-1$ or $t-2$), but rather trials $t-4$ and $t-5$ (Figure 3C; trial $t-4$, mean difference = 0.117, $p < 0.05$, $t(27) = 2.09$; trial $t-5 = 0.170$, $p < 0.01$, $t(27) = 2.84$).

The view that omitted rewards only evoke exploration when they coincide with particular phases of sympathetic arousal also implies that the onset of exploration should be phase-locked to the Mayer wave frequency (see Methods). The median trial duration was ~3 seconds (range = [2.2, 3.2]), so a ~5-trial cycle would correspond to a 0.06–0.09 Hz oscillation, which aligns with the frequency range of the Mayer wave. We found that pupil phase at the onset of exploration was concentrated at the rising phase (Figure 3D; mean phase = 47.18° , Hodges-Ajne test, $p < 0.01$; vector length = 0.075; null = 0.026, 95% CI = 0.004 to 0.057, $p < 0.0001$). In contrast, pupil phases during reward-matched trials pointed in the opposite direction (mean phase = 207.58° ; significantly different from onsets, $p < 0.02$, Watson's $U^2 = 0.25$, $n = 2170$ phases including 1135 onsets). Together, these results support the hypothesis (Figure 3B) that slow, rhythmic fluctuations in arousal interact with reward history to determine the timing of exploration onset.

The revised figure panel is as follows:

Reviewer #2 (Remarks to the Author):

In this study, the authors used a three-armed bandit task to examine the role of pupil dilation in predicting transitions between exploratory and exploitative internal states, as estimated by a Hidden Markov Model. They further investigated how pupil activity influenced frontal eye field (FEF) neuron activity driving choice-related saccades.

The authors found that exploration states were associated with larger pupil size compared to exploitation states, and that the relationship between pupil size and the probability of exploring followed a U-shaped function. Pupil size ramped up across trials preceding the onset of an exploration state and began shrinking immediately afterward, suggesting that pupil dilation plays a role in initiating—but not maintaining—exploration. This dynamic may explain its U-shaped relationship with exploration probability. Control analyses indicated that reward history might contribute to this ramping effect, though an offset remained specific to trials preceding exploration states. Further analyses suggested that the link between larger pupil size and exploration states could be driven by both reward history and arousal fluctuations over time.

At the neural level, larger pupil size was associated with increased disorganization in FEF neuronal choice encoding, and this disorganization mediated the pupil's role in triggering exploration. Additionally, pupil size influenced both neural speed and response times, suggesting a role in decision-making slowing at both behavioral and neural levels.

Overall, the study is well-structured, addresses important questions, and provides novel insights into the relationship between pupil size, neural activity, and exploration. However, I have some concerns regarding the organization of the results and the clarity of certain aspects.

We appreciate the reviewer's comments on the strengths of this work as well as their extremely careful read of our manuscript and detailed feedback. We appreciate the time and energy that was put into this review. We hope this revision addresses these important concerns around organization and clarity.

MAJOR COMMENTS

Results

- I find somewhat difficult to follow the narrative while reading the Results section. While the logical progression aligns with the analysis workflow, it makes it challenging for the reader to grasp and integrate key takeaways throughout. For example, by the time the reader reaches line 302, the

initial conclusion—that pupil ramping directly predicts the onset of exploration—is reconsidered in light of control analyses suggesting a role for time and reward history. This leads to a shift in focus toward further analyses. While the control analyses are highly valuable, they seem to diverge from the central message, making the narrative feel overwhelming and somewhat distracting.

Consider streamlining the Results section to maintain a clearer emphasis on the main findings while still incorporating the necessary control analyses in a way that enhances, rather than diffuses, the key message.

We thank the reviewer for this insightful comment and appreciate how the large number of controls distracted from the main points of the manuscript. To address this, we revised the Results section to enhance clarity and guide the reader more effectively through the key takeaways. Specifically:

1) We added descriptive subheadings throughout the Results to emphasize the main conclusions of each section (e.g., *“Pupil size ramps up before exploration and shrinks after its onset”* and *“Neural disorganization mediates the relationship between pupil size and exploration”*). These subheadings help highlight how each set of results builds to the central conclusion.

2) We revised the title of the paper in response to a suggestion from another reviewer, but believe this change also addresses this concern because it simply and succinctly states the central conclusion of the manuscript: *“Pupil size predicts exploration through critical slowing in prefrontal dynamics”*.

3) We revised transitions and paragraph endings to clearly signal how control analyses (e.g., reward-matched comparisons, structural equation models) serve to support—not reconsider—the initial conclusion that pupil-linked arousal predicts the onset of exploration. For instance, when addressing potential confounds like reward history or time, we now explicitly state that these effects do not fully account for pupil ramping.

4) We clarified the purpose of each result section, so that the Results section now reads as a cohesive narrative arc: from observing a robust link between pupil size and the onset of exploration, to systematically ruling out alternative explanations, and finally to identifying neural mechanisms that mediate this relationship.

We believe that this revised structure preserves analytical rigor while substantially improving clarity and readability. As these revisions affect the entire Results section, we have highlighted the changes directly in the manuscript.

- [Lines 184-202] The rationale behind the median split analyses needs clarification. Specifically, it would be helpful to explain why the authors selected reward, peak velocity, scatter index, and learning index for their control analyses and how targeting these dimensions helps exclude disengagement effects.

The previous version of this manuscript did not motivate the specific behaviors we used as controls here. This was an oversight—each of these measures is one we have previously shown differentiates exploration from exploitation in prior work (Chen et al., 2021; Ebitz et al., 2018; Laurie et al., 2025). They were chosen simply because they are the standard diagnostic behaviors in our lab. We have now explained this point in the Results section on page 8:

[...] if this were the case, then the valid, large-pupil explore choices would systematically differ from the false, small-pupil “explore” choices. To evaluate this possibility, we compared small- and large-pupil explore trials across 5 key behavioral and neural dimensions that we have previously shown differentiate exploration from exploitation (Chen et al., 2021; Ebitz et al., 2018; Laurie et al., 2025): reward rate, saccade velocity, the neural scatter index, a trial-wise learning index, and reaction time. Small- and large-pupil explore choices (median split) were essentially indistinguishable along each of the 5 key dimensions (Figure 1F).

- In general, authors refer to a “baseline” all along the results section, but it is not clear what they mean by baseline. It seems to correspond to $z\text{-score} = 0$, but this should be explicitly stated to facilitate reading. Additionally, this information is missing from the Methods section and should be included.

The word baseline was used to refer to the specific baseline of each analysis, but we understand that this was unclear. Where it was possible to remove the word baseline and replace it with a more precise phrase, we have done so. Now we only use the term baseline to refer to the average pupil size across trials ($z\text{-score} = 0$) and define the term twice: first, when it first appears in the results (on page 8 in line 222) and then in the Methods (Pupil preprocessing; page 26).

Added to the Results:

Here, "baseline" refers to the average pupil size across trials (i.e. z-scored value of 0; see Methods).

Added to the Methods:

In the results section, we refer to a z-score of 0 as "baseline".

• Authors should consider adding labels to the task procedure in Figure 1A, with information about the neuronal recording period as well. Also, clarify the meaning of T1, T2, and T3 in this figure, as this only becomes explicitly clear in Figure 4.

We thank the reviewer for this helpful suggestion. We have added labels (T1, T2, and T3) to the targets in Figure 1A to clarify their identities. Placing these labels directly next to each target in the first panel makes it immediately clear which target is which. This improves figure readability and helping readers better understand the task structure. Additionally, to clarify the period of neural responses used for analysis, we have added a description of the neural analysis epoch to the Methods section (page 26).

In the results section, we refer to a z-score of 0 as "baseline". The 200 ms window immediately preceding target onset was chosen as the analysis epoch for all choice-predictive neural measures and firing rates were averaged over this epoch within trials. A longer, whole-trial epoch was chosen for neural speed analyses (0 to 400 ms following target presentation; see below).

Figure 1. Task design and pupil. A) Top: Subjects made saccadic choices between three identical options (T1, T2, and T3). One of the options (e.g., T1 in this example trial) was located in the receptive field of **a neuron in the frontal eye field (FEF; dotted circle)**. Bottom: Reward probabilities for the 3 options (lines), with choices overlaid (dots) for 200 example trials. Gray bars = explore-labels.

- In Figure 1C It is unclear what type of beta weights are reported. Are these from a GLM, an HMM/RL model, or something else? Please specify in the figure description and/or the Methods section.

This was an oversight, and we've added detail of the procedure used to generate the beta weights in Figure 1C to the figure caption and methods. Briefly, these were derived from generalized linear models (GLMs) in which scatter index and response time were modeled as dependent variables, and explore state (0 = exploit, 1 = explore) was used as the independent variable. The regressions were fit separately for the explore-state labels derived from the HMM and from a reinforcement learning (RL) model. This is now explained by the following changes to the manuscript:

Caption of Figure 1C:

C) Comparison of regression coefficients for HMM-inferred and RL-inferred explore choices, predicting either the disorganization of neural population responses ("scatter index"; see Methods; Ebitz et al., 2018) or response time. **Separate models were fit using either the explore labels from the HMM or from an RL model.**

These GLMs are now described in the methods:

Generalized Linear Model (GLM). To examine the relationship between behavioral and neural variables, we employed generalized linear models (GLMs). These models were fit using maximum likelihood estimation. The dependent variable Y_i (e.g., scatter index, response time, or choice probability) was modeled as a linear combination of predictors:

$$Y_i = \beta_0 + \beta_1 \cdot \text{explore state}_i + \beta_2 \cdot \text{pupile size}_i + \beta_3 \cdot (\text{explore state}_i \times \text{pupile size}_i) + \varepsilon_i$$

where Y_i is the dependent variable on trial i , B_0 is the intercept, and B_1 , B_2 , B_3 are regression coefficients quantifying the influence of the corresponding predictors (e.g., explore state, pupil size, and their interactions). ϵ_i is the residual error term.

In analyses where categorical variables (e.g., explore state: 0 = exploit, 1 = explore) were used as predictors, these were coded as binary dummy variables. Models were fit using the identity link function and assumed normally distributed errors.

We also realized that insufficient detail of the Hidden Markov Model (HMM) was included in the previous manuscript. This has been added on pages 26-27:

Hidden Markov Model. To identify when subjects were exploring versus exploiting, we employed a hidden Markov model (HMM; Ebitz et al., 2018, 2020; Chen et al., 2021). In this framework, choices Y_t are treated as emissions from a latent decision-making state z_t , which can either be an explore or one of multiple exploit states.

The emission model for exploration assumed a uniform probability of selecting any option:

$$p(y_t = k | z_t = \text{explore}) = \frac{1}{N_k}$$

where N_k is the total number of options. In contrast, exploit states deterministically emitted choices of the exploited option i :

$$p(y_t = k | z_t = \text{exploit}_i, k = i) = 1$$

$$p(y_t = k | z_t = \text{exploit}_i, k \neq i) = 0$$

Latent state transitions followed a Markov process, such that the probability of the current state depended only on the previous state:

$$p(z_t | z_{t-1}, y_{t-1}, \dots, z_1, y_1) = p(z_t | z_{t-1})$$

This process is described by a time-invariant transition matrix defining the probability of transitioning between all possible state pairs:

To reduce model complexity, parameters were shared across exploit states, and subjects were assumed to begin in the explore state. The final HMM included only two free parameters: the probability of persisting in exploration and the probability of persisting in exploitation. The model was fit using expectation-maximization (Baum-Welch algorithm; Bilmes, 1998) with 20 random restarts, and the solution maximizing the observed data log-likelihood was selected. The most probable sequence of latent states was recovered using the Viterbi algorithm.

- The meaning of the dotted line in Fig 4B is unclear. Please explain it in both the text and the figure legend.

The dotted line in Figure 4B indicates the expected number of neurons that would be significant by chance at a $p < 0.05$ threshold, based on the total number of tested neurons (i.e., the false-positive rate under the null hypothesis). This line serves as a reference for interpreting the observed counts of significant neurons. This is clarified in the figure legend:

The dashed line indicates the number of neurons expected to be significant by chance at $p < 0.05$.

- Figure 4D: The term “All Tuned Neurons” is ambiguous. Clarify whether this refers to 88 out of 155 neurons or 21 out of 88.

In Figure 4D, “All Tuned Neurons” refers to the 88 neurons out of 155 that were significantly tuned for choice. This group includes neurons regardless of whether they were modulated by pupil size. We have now clarified this in both the figure legend and in the Results.

Caption of Figure 4D:

Same for all tuned neurons, which refers to the 88 out of 155 FEF neurons that were significantly tuned for choice, regardless of their modulation by pupil size.

In the Results section on page 15:

Out of 155 recorded single neurons, 88 (57%) were tuned for choice (Figure 4B; 57%, one sample proportion test: $p < 0.001$). **These are referred to as "tuned neurons," regardless of whether they were modulated by pupil size.**

- Figures 4F and 4G: The main panels show all trials, while the insets depict exploit-only trials. Given that the study focuses on explore trials, it would be more relevant to show explore-only trials in the insets for both decoding and the scatter index.

The exploit-only insets in Figures 4F and 4G demonstrate that pupil size predicts variability in decoding accuracy and neural scatter even outside of exploratory states. This analysis supports our central claim that pupil-linked arousal modulates the structure of neural population activity more broadly, not just during exploration—a point we now clarify in the discussion (see below). However, we appreciate the reviewer's point that it may also be interesting to know whether pupil size predicts these measures within periods of exploration. We have now also generated the same analyses for explore-only trials and included these results in the supplementary **Figure S2**. Given that our overarching hypothesis was that pupil size predicts the onset of exploration, rather than modulating neural variability within exploration, it is perhaps not surprising that we did not see any effect here.

This is now described in the Results on page 17, lines 463-468 for choice decoding

This was not driven by differences between the states because pupil size also predicted choice decoding accuracy within exploit trials alone (GLM: $\beta = -0.003$, $p < 0.05$, 95% CI = -0.006 to -0.001). There was no significant effect of pupil size on choice decoding within explore trials (GLM: $\beta = -0.0009$, $p = 0.83$, 95% CI = -0.01 to 0.008 , Figure S2B), though decoding accuracy was already close to chance in these trials.

and on page 17, lines 475-482 for the scatter index.

We observed a higher scatter index during exploration compared to exploitation (paired t-test: both subjects, $p < 0.0001$, Figure S2C). Here, we also found that increasing pupil size predicted an increase in the scatter index (Figure 4G; GLM: $\beta = 0.006$, $p < 0.0001$, 95% CI = 0.004 to 0.009). This effect remained significant and of similar magnitude within exploit trials alone (GLM: $\beta = 0.004$, $p < 0.005$, 95% CI = 0.002 to 0.007), again suggesting that the relationship between pupil size and scatter was not an artifact of state differences

with pupil size. Pupil size again did not significantly predict the scatter index during explore trials (GLM: $\beta = -0.0003$, $p = 0.93$, 95% CI = -0.009 to 0.0008 , Figure S2D).

Supplemental Figure 2:

Figure S2. Decoded choice probability and scatter index across behavioral states and pupil size. (A) Decoded choice probability (projection onto the correct coding dimension) for exploit and explore states. Dots represent individual sessions, with lines connecting values from the same session across states. **(B)** Decoded choice probability plotted as a function of pupil size quantile for explore trials alone, related to Figure 4F. **(C)** Scatter index, a measure of variance in choice-predictive population activity, for exploit and explore states, with lines connecting values from the same session across states. **(D)** The scatter index plotted as a function of pupil size quantile for explore trials, related to Figure 4G.

The relevant passage from the discussion:

Notably, pupil size also predicted slower neural and behavioral responses within exploit-only trials. This suggests that these effects are not an artifact of differences between explore and exploit states, but this result also makes it impossible to interpret these patterns of neural activity as specific signatures of the onset of exploration itself. Instead, it is more likely that arousal has a domain-general influence on neural dynamics, causing ongoing fluctuations in information encoding and speed. In this view, the transition into exploration in FEF may best be thought of as the result of a critical tipping point that occurs when the conditions necessary for exploration to occur (i.e., domain-general fluctuations neural dynamics linked to arousal) happen to coincide with evidence in favor of exploration (i.e., sequences of omitted rewards).

- [Line 387] Since the authors report a decrease in decoded choice probability during explore choices compared to exploit choices, it would be informative to include a plot directly comparing these two conditions.

To address this comment, we have added panels to Figure S2 that directly compares decoded choice probability and scatter between explore and exploit states.

Figure S2. Decoded choice probability and scatter index across behavioral states and pupil size. (A) Decoded choice probability (projection onto the correct coding dimension) for exploit and explore states. Dots represent individual sessions, with lines connecting values from the same session across states. (B) Decoded choice probability plotted as a function of pupil size quantile for explore trials alone, related to Figure 4F. (C) Scatter index, a measure of variance in choice-predictive population activity, for exploit and explore states, with lines connecting values from the same session across states. (D) The scatter index plotted as a function of pupil size quantile for explore trials, related to Figure 4G.

- Figures 4J-5C-F/Tables of Supplementary: There are inconsistencies in reporting the total effect (c). In Figure 5C, effect values (c, a, b, c') are rounded to the second decimal place, while other figures and tables use three decimals. Additionally, the total effect is reported as 0.090 in Figure 4J, 0.09 in Figure 5C, and 0.095 in Figure 5F. If these values are derived from identical analyses, the discrepancy should be explained. If they reflect different analyses, this distinction should be made clear.

We thank the reviewer for catching this inconsistency. We agree that the total effect in the mediation analyses in Figures 4J, 5C, and 5F should, in theory, match. However, the small differences we observe in practice are due to variations in the trials included in each analysis. These differences arise because the mediators (RT, speed, and scatter) are calculated differently

and contain different patterns of missing data, as well as because outliers in neural activity and behavior do not necessarily occur on the same trials.

For example, RT slowing is calculated as a difference between adjacent trials, which means trials with missing or invalid RT values (e.g., due to aborts or extreme outliers) are excluded from the analysis. As a result, the mediation analyses using RT as the mediator included 21,765 trials, while those using speed and scatter as mediators included 21,535 and 21,764 trials, respectively. These slight differences in trial inclusion propagate into the total effect estimates. To address this point, we added a row to the table that provides a full report for each analysis, including the number of sessions and trials for each.

All values in the figures and tables are now reported to three decimal places for consistency. Previously, some values were rounded to two decimal places where the third decimal was zero, but we appreciate the value of precision here.

Materials and Method

- It is not clear if the animals were trained before the experiment and how. If this was the case, as I assume, I would recommend that the authors introduce training information in the Materials and Method section.

We have added training information to the Methods on page 25:

In order to train the animals on the explore/exploit task a gradual procedure was used in which the two animals were first trained to make saccadic eye movements in exchange for liquid rewards. Once the animals reliably made controlled eye movements to a single target (generally within 1–2 days), a second target was introduced, and the animals were free to choose between them. At the outset, each target was associated with a probability of reward (initially 10% and 90%), which was reversed in blocks at the experimenter’s discretion. Over a period of 2–4 months, the difference in reward probabilities between the targets was gradually reduced, the blocks transitioned into gradual reward probability shifts (reward walks), and a third target was introduced. The speed and order of these changes depended on each animal’s performance and engagement with the task. One animal (monkey O) was naïve to laboratory tasks prior to this experiment, whereas the

second (monkey B) had been previously trained on covert and overt attention tasks, but not on any prior value-based tasks.

MINOR COMMENTS

- [Line 146] – The acronym "FEF" appears here for the first time. For clarity, spell it out in full before introducing the abbreviation.

We now spell out the phrase before we introduce the acronym in both the caption of Figure 1A and the main text.

- [Line 175-176]: The phrase "there was also a U-shaped relationship between pupil size and the probability of switching" is ambiguous. Clarify whether "switching" refers to a switch choice or a state switch.

We have clarified that we mean decisions that differ from the previous decision:

In order to determine whether this pattern was also apparent in raw switching probability (i.e. not the HMM-model labels), we next asked if pupil size predicted choices to a different option than the previous trial.

- Figure 1F The text refers to a "blue line" [Line 157], but no blue line appears in the figure. Additionally, the panel order in Figure 1F does not match the order described in the text. Consider placing the "learning index" as the last panel to maintain consistency.

The reference to the "blue line" was from an earlier version of the figure, where the line was gray in the version we submitted. We have corrected the text to refer to the "gray line".

We have also reordered the panels in Figure 1F to match the sequence in the main text (reward rate, saccade velocity, neural scatter, and learning index), with a new panel for reaction time at the end, as requested by another reviewer.

- [Lines 224-225] Specify which neural measures are being referred to in this section.

We have rephrased this sentence to make it clearer that it refers to a previously published paper.

This null result resonates with results seen in neural data in prior work (Ebitz et al., 2018).

- Figure 4H-I-J There is a mismatch between panel labels and their descriptions in the legend:

- o Panel H corresponds to the Figure 4I legend.

- o Panel I corresponds to the Figure 4J legend.

- o Panel J corresponds to the Figure 4H legend.

Please update the figure legend accordingly.

We have now corrected the figure legend so that each panel (H, I, J) matches its correct description.

- [Lines 399-400] The phrase "both subjects" in "we observed that increasing pupil size predicted an increase in the scatter index in both subjects (Figure 4G; GLM: $\beta = 0.04$, $p < 0.0001$)" is unclear. Specify which subjects are being referred to.

We've removed this phrase as it seemed to create unnecessary confusion. It was just being used here to mean "in all the data."

- [Line 451-455] – The results for neural speed are different compared to those of response times, but the statements accompanying stats read ambiguous. Specifically, for response times, the authors report that "Response time was not only slower in the trials before exploration [...] but it slowed down over trials before the onset of exploration". For neural speed, they say that "Like response time, neural speed was also significantly slowed in the trials before exploration, compared to matched-reward controls". Since the interaction term is not significant for neural speed, this distinction should be explicitly stated to avoid ambiguity.

We have revised the relevant sentence in the Results on page 20 in lines 536-539 to clarify the distinctions in the interaction terms between the two results:

Like response time, neural speed was also significantly slower on average in the trials before exploration, compared to matched-reward controls (Figure 5D–F; GLM offset = -

0.17, $p < 0.0001$, $n = 28$). However, unlike response time, neural speed did not show a significant interaction ($\beta = -0.01$, $p = 0.08$).

- Since no figures are referenced in the Discussion except Figure 1C, consider removing this mention for consistency.

We have removed the reference to Figure 1C from this section.

Typos

[Line 448]: missing bold font for Figure 5A-C.

[Line 496]: “latent explore and explore states” should be corrected in “latent explore and exploit states”.

[Line 385]: A double closing parenthesis is present, please correct it.

Have all been corrected in the manuscript. Thank you for the detailed reading.

Reviewer #3 (Remarks to the Author):

The authors investigated the role of pupil size in exploration behavior and its underlying neural mechanisms using simultaneous recordings of pupil size and frontal eye field (FEF) activity in an explore/exploit task in two behaving monkeys. They show a U-shaped relationship between tonic pupil size and exploration probability. Moreover, pupil size linearly correlates with the onset of exploration, and this effect remains pronounced even after controlling for reward history. More importantly, pupil size is linked to FEF activity and saccadic reaction times. Overall, the manuscript is very well-written, the experiments are well-designed, and the results are highly compelling. These findings contribute significantly to the literature on exploration-exploitation behavior, demonstrating that pupil-linked mechanisms mediate the onset of exploration by altering prefrontal cortex activity.

We appreciate the reviewer's kind words about the design, execution, and contributions of our work. We hope the following changes address their outstanding concerns with the manuscript.

1) The main limitation is that “neuronal tuning functions are too noisy to partial out the contributions of exploration and pupil size,” preventing detailed analyses of the relationship between pupil size and FEF activity in the context of exploration behavior. Beyond this, I have only a few relatively minor comments.

Although we were not able to independently estimate the contribution of choice, explore/exploit state, and pupil size at the level of single neurons, we were able to differentiate these factors at the neural population level because we were able to get reasonable single-trial measures of neural activity through combining across simultaneously recorded units. Briefly, we found that both of our population measures of choice selectivity—i.e. decoding accuracy and the scatter index—were sensitive to both explore/exploit state and pupil size. These converging results strengthen our conclusion that pupil size predicts a reduction in choice-predictive structure in FEF activity—even during periods of exploitation—supporting our interpretation of pupil-linked arousal as a driver of network-level changes.

These results appear in Figures 4 and 5.

2) The authors argue that the U-shaped (instead of linear) relationship observed between pupil size and exploration choice probability (Fig. 1E) is driven by pupil size from non-first exploration trials. While some indirect evidence is provided (e.g., Fig. 2D), the authors do not present direct evidence to support this argument. It is suggested that they plot Fig. 1E to analyze the probability

of non-first exploration as a function of pupil size. According to their hypothesis, a smaller pupil size should be associated with a higher exploration probability.

This is a great suggestion and we've added this analysis to the revised manuscript. To be clear, our hypothesis is not that the relationship should flip (i.e. that smaller pupil size would now predict more exploration for the explore trials that are not onsets). Instead, because pupil size gets smaller across multiple explore trials, we would predict that excluding the onsets should weaken, if not eliminate, the positive, linear relationship. Indeed, this is what we found when performing the same analysis as Figure 2E, but excluded the onsets: the slope was reduced and the U-shaped relationship was still apparent in the combined monkey data.

We have added this analysis to the manuscript as supplement Figure S1 and have updated the Results text accordingly to describe the finding on pages 12-13. This result reinforces the interpretation that the U-shaped curve in Figure 1E arises from the combination of high pupil size predicting the onset of exploration and low pupil size being more common during ongoing exploratory states. In the results:

If the U-shaped relationship between pupil size and exploration (Figure 1E) were driven primarily by later explore trials, it should remain evident after excluding onset trials from the analysis. Moreover, removing onsets should substantially reduce the slope of the linear effect. To test this, we repeated the analysis using only later explore trials. As expected, the linear slope decreased in both subjects (Figure S1): for subject B, β_1 dropped from 0.063 (all explore trials) to 0.018 (excluding onsets), and for subject O, from 0.110 to 0.075. In contrast, the quadratic terms remained relatively stable: for subject B, β_2 was 0.091 for all explore trials and 0.041 with onsets excluded; for subject O, β_2 was 0.240 and 0.215, respectively. Importantly, the U-shaped relationship persisted when data were combined across both subjects (Figure S1), with the quadratic model significantly outperforming the linear model (AIC quadratic = -1108.84, linear = -1103.30; relative AIC weight for the quadratic model = 0.941). These findings confirm that the nonlinearity observed in the original analysis (Figure 1E) was driven by the decrease in pupil size in later exploratory trials, whereas the onset of exploration had a largely linear relationship with pupil size.

And the new figure panel appears as follows:

Figure S1. Same as Figure 1E, but excluding first explore trials.

3) One of the main findings of the manuscript is the linear relationship between pupil size and exploration onset (Fig. 2E). It is therefore important to present individual monkey data to confirm that both monkeys exhibit a similar pattern of results.

To confirm that the linear relationship between pupil size and the onset of exploration (Figure 2E) is present in both animals, we have added subject-specific panels to Figure 2E. These analyses show that the linear relationship holds independently for both subject B and subject O, consistent with the pooled results. We have also updated the Results section to reflect this change on page 12.

Indeed, pupil size had a primarily linear relationship with the onset of exploration **in both subjects** (Figure 2E; 1st order GLM: $\beta = 0.042$, $p < 0.0001$, 95% CI = 0.031 to 0.053, AIC = -1973.57, $n = 28$). Adding a quadratic term did not substantially improve the model fit ($\beta_2 = 0.038$, $p = 0.073$, 95% CI = 0.003 to 0.080; quadratic model AIC = -1974.81; Δ AIC = -1.24; AIC weight of quadratic model = 0.65; see Methods). This linear relationship was also observed in both monkeys individually (see Figure 2E, right panels). For subject B, a first order GLM confirmed a significant positive association ($\beta = 0.044$, $p < 0.0001$, 95% CI = 0.028 to 0.060, AIC = -699.13, $n = 10$), and adding a quadratic term did not improve the model fit ($\beta_2 = 0.049$, $p = 0.0878$, 95% CI = 0.006 to 0.104; Δ AIC = -0.96). Similarly, for subject O, pupil size showed a significant linear relationship with exploration onset ($\beta = 0.034$, $p < 0.0001$, 95%

CI = 0.019 to 0.049, AIC = -460.29, n = 18), and the quadratic model again provided no additional explanatory power ($\beta_2 = 0.022$, $p = 0.402$, 95% CI = -0.030 to 0.075; $\Delta AIC = +1.30$). These results confirm that the linear relationship between pupil size and the onset of exploration was robust across both subjects and not driven by outliers or subject-specific variability. Conversely, there was no special relationship between pupil size and probability of starting to exploit (1st order GLM: $\beta = 0.05$, $p > 0.05$). Thus, pupil size specifically predicted the onset of exploration, rather than explore choices or state switches more generally.

Caption of Figure 2:

E) The probability of starting to explore as a function of pupil size quantile. Solid line: Linear GLM fit. Error bars and shaded regions depict mean \pm SEM. **Insets: Same analysis shown separately for each monkey.**

4) "A total of 88 out of 155 single neurons were tuned for choice, and of those, 21 were also modulated by pupil size, while 16 showed a significant interaction between choice and pupil size." This indicates that 43% of FEF neurons are not modulated by choice. This raises an interesting question: Are these 43% of FEF neurons modulated by pupil size? If some portion of these "task-independent" FEF neurons is influenced by pupil size, would the same effects be observed in the relationship between pupil size and neural speed (Fig. 5E)? If so, this may suggest that pupil-linked mechanisms operate in a more domain-general manner, affecting not only exploration-exploitation behavior but other types of behavior as well.

We appreciate the reviewer's thoughtful point and now examine whether pupil-linked effects are present in FEF neurons that are not modulated by choice. We identified non-choice-tuned neurons and tested whether pupil size predicted trial-by-trial firing rate using a GLM. Out of 155 neurons, 67 (43%) were not significantly tuned for choice. Among these, 22% (15/67) showed a significant relationship between pupil size and firing rate ($p < 0.05$), with a median pupil effect (β) of -0.0011 . These findings suggest that pupil-linked mechanisms influence FEF activity even in neurons that are not directly involved in encoding the current choice, consistent with a broader, possibly domain-general role for arousal in shaping neural dynamics, as suggested by the reviewer. We have added to the result section on page 15 to address this reviewer point.

Among untuned neurons, an additional 22% (15/67) were significantly modulated by pupil size ($p < 0.05$), with a median regression coefficient (β) of -0.0011 ± 0.063 . This suggests that pupil-linked mechanisms affect FEF activity even in neurons that are not directly involved in encoding choice. This may suggest a more domain-general role for arousal in modulating prefrontal network dynamics.

5) Similar effects were observed in exploit-only trials (Fig. 5B and 5E), suggesting that the link between pupil size and reaction time (or neural speed) may represent a more general effect, independent of the exploration-exploitation context.

This is an important observation that is worth folding into our discussion. Here are those additions, Discussion, page 23:

Notably, pupil size also predicted slower neural and behavioral responses within exploit-only trials. This suggests that these effects are not an artifact of differences between explore and exploit states, but this result also makes it impossible to interpret these patterns of neural activity as specific signatures of the onset of exploration itself. Instead, it is more likely that arousal has a domain-general influence on neural dynamics, causing ongoing fluctuations in information encoding and speed. In this view, the transition into exploration in FEF may best be thought of as the result of a critical tipping point that occurs when the conditions necessary for exploration to occur (i.e., domain-general fluctuations neural dynamics linked to arousal) happen to coincide with evidence in favor of exploration (i.e., sequences of omitted rewards).